# Inhibiting translation elongation by reducing eIF5A activity induces feedback inhibition of initiation, limiting tumour cell proliferation

Cancer development is associated with dysregulation of the translatome, and targeting canonical eukaryotic initiation and elongation factors can offer treatment avenues for various neoplasms. Emerging evidence indicates that dysregulated mRNA elongation, involving alterations in eEF2 activity and eIF5A expression, also contributes to tumour cell growth. In this study, we investigate whether targeting eIF5A with the inhibitor GC7 is a viable strategy to curtail aberrant cell growth. Our findings demonstrate that inhibiting elongation by reducing eIF5A activity induces feedback inhibition of initiation through eIF2α phosphorylation, decreasing ternary complex formation and shutting down bulk protein synthesis. Employing dynamic SILAC, we identify proteins impacted by reduced eIF5A activity, and show their decreased translation results from feedback inhibition to initiation or other processes downstream of eIF5A. Decreased eIF5A activity impairs mitochondrial function, which activates signalling through HRI to eIF2α phosphorylation, reducing cancer cell proliferation. These effects are reversed by treatment with the integrated stress response inhibitor, implying that the impact of GC7 on cancer cell proliferation is mediated via translation initiation rather than elongation inhibition. These data suggest that eIF5A inhibition could be used to target cancer cells that depend on mitochondrial function for their proliferation and survival.

Neoplasia is associated with a dysregulated translatome, and it is recognised that cancerous cells rewire protein synthesis to adapt to the adverse conditions within the tumour microenvironment[1]. Novel therapeutic interventions under development that target protein synthesis for the most part focus on inhibition of selected eukaryotic initiation factors e.g. eIF4A1[2]. However, the data show that the elongation stage of protein synthesis also represents a major regulatory node[2]. For example, in colon cancers associated with adenomatous polyposis coli (APC) deletion, increases in protein synthesis occur through activation of translation elongation and not initiation, mediated by increased eEF2 activity[3]. Elevated expression of the elongation

factor eIF5A is also associated with the development of several types of cancer[4,5], including lung metastasis[6] and eIF5A is a prognostic marker for poor lung adenocarcinoma patient outcome, however a lack of understanding of its role in mRNA translation restricts therapeutic targeting strategies[7–10].

The two isoforms of eIF5A (1 and 2) are the only proteins known to undergo the post-translational modification hypusination. This process is orchestrated by DHPS and DOHH, which add the epsilon aminobutyl moiety of spermidine to lysine 51 of eIF5A and perform the final hydroxylation of the molecule, respectively[11–14]. Upon activation, eIF5A binds to the E-site of the ribosome, projecting the

e-mail: arissfak92@gmail.com; rfh32@mrc-tox.cam.ac.uk; aew80@mrc-tox.cam.ac.uk

hypusine-containing domain towards the P-site. This positioning allosterically facilitates peptide bonds formation[15]. The degree of specific peptide requirement for eIF5A function for translation elongation appears to be related to the system studied[4,16,17]. For example, data from yeast and in vitro assays, strongly suggest that di- and tri-peptide sequences that are enriched for prolines have a high requirement for eIF5A[16,18,19]. In mammalian cells it is less clear. There are examples of eIF5A impacting on the synthesis of selected proteins e.g. c-Myc, however, global quantitative proteomics approaches have not led to the identification of specific peptide motifs, suggesting its function is more nuanced in higher eukaryotes[20–24].

Studies show that inactivation eIF5A or a reduction in its protein levels using siRNA-based approaches or knockout strategies results in mitochondrial dysfunction in fission yeast, MEFs, macrophages and mice[4,16,17,22]. This mitochondrial impact could be related to the toxicity observed in pre-clinical studies of the eIF5A inhibitor GC7 which restricted its clinical use[24]. However, providing such toxicities are understood and managed, selective targeting of mitochondrial complexes can provide an effective strategy to treat cancers that are highly dependent on oxidative phosphorylation (OXPHOS) for their survival[25]. For example, chemotherapy-resistant AMLs show an OXPHOS dependency that can be directly targeted via inhibition of mitochondrial function[26].

To determine the utility of targeting eIF5A as an anti-cancer therapeutic we analysed the temporal changes in mRNA translation following treatment with GC7. We identified that a decrease in eIF5A activity resulted in a sequential inhibition of translation elongation and then translation initiation. We used Dynamic SILAC to understand the likely protein targets downstream of eIF5A, and our data show that despite the identification of ribosomal, translation, and mitochondrial proteins, as previously described[22], there was no evidence for eIF5A-dependent peptide motifs. We hypothesised that the observed effects likely stemmed from broader disruptions in translation or other processes affected by eIF5A inhibition and used ISRIB, an integrated stress response (ISR) inhibitor[27], to identify targets particularly sensitive to eIF2α phosphorylation. We identify the mechanistic link between eIF5A inhibition and eIF2α phosphorylation and show that inhibition of eIF5A activity triggers mitochondrial dysfunction, activating mitochondrial proteases OMA1 and DELE1[28], which in turn activate the eIF2α kinase, HRI, leading to inhibition of translation initiation via phosphorylation of eIF2α. We examined the impact of GC7 in a panel of control and tumour-derived cell lines and identify a link between functionality of mitochondria and sensitivity to GC7. Moreover, we found that mitochondrial stress sensed through OMA1-DELE1-HRI inhibits cancer cell proliferation thorough a P-eIF2α-dependent mechanism.

Taken together, these data suggest that cancer cells with minimal reserve mitochondria capacity, or tumours which are dependent on OXPHOS for survival and growth, are more sensitive to eIF5A inhibition and that devising therapeutic interventions that target this protein could provide a viable therapeutic option.

## Results

### Inhibition of hypusination leads to a sequential inhibition of translation elongation and initiation

Previous studies on eIF5A have predominantly examined downstream changes after prolonged periods of eIF5A inhibition e.g., 24 h. To comprehensively explore mRNA translation dynamics upon eIF5A inhibition, we assessed the impact of the eIF5A hypusination inhibitor, GC7, over a time course, from 30 min to 24 h. As high levels of eIF5A are a prognostic marker for poor lung adenocarcinoma patient outcome, we chose to study the effects of GC7 in A549 cells and observed a significant reduction in hypusination within 3 h (Fig. 1a, b). We used an antibody specific to eIF5A1 as it is the predominant isoform expressed in A549 cells (Supplementary Fig. 1a), and observed a small

reduction in expression after 24 h (Fig. 1c), in line with previous observations that hypusination stabilizes eIF5A[13]. The decrease in hypusination at 3 h correlated with a significant increase in p-eEF2, a recognized marker of translation elongation inhibition (Fig. 1d). To evaluate translation initiation, levels of p-eIF2α and p-4EBP1 were assessed. P-eIF2α levels significantly increased from 6 h (Fig. 1a, e) and correlated with a significant increase in ATF4 expression (Fig. 1a, f). Conversely, phosphorylation of 4EBP1 was not significantly diminished until 18 h, indicating that the inhibition of mTOR signalling did not occur during the early stages of eIF5A inhibition (Supplementary Fig. 1b, c). Taken together, these data suggest that following inactivation of eIF5A there is a correlation between the inhibition of elongation followed by the inhibition of initiation.

To confirm that the induction of ATF4 expression was dependent on p-eIF2α observed from 6 h, we utilised mouse embryonic fibroblasts (MEFs) harbouring an alanine substitution at position 51 (S51A) to prevent phosphorylation of eIF2α. We treated S51A and WT MEFs with GC7 for 24 h which decreased hypusine and increased p-eEF2 levels (Supplementary Fig. 1f–i). Although WT MEFs significantly increased ATF4 expression after treatment with GC7, this response was absent in S51A MEFs, suggesting the induction of ATF4 was dependent on p-eIF2α (Supplementary Fig. 1f–k).

To understand the effects of eIF5A inhibition on protein synthesis, a puromycin incorporation assay was used (Fig. 1g, h). Western blot analysis revealed an increase of puromycin incorporation during the early stages of GC7 treatments (0.5 h). As puromycin incorporates into the A-sites of ribosomes, this increase may reflect the accumulation of slow-moving ribosomes on mRNAs, in line with the observed increase in p-eEF2, rather than increased translation rates. However, there was a decrease in puromycin incorporation from 18 h (Fig. 1g, h) which is likely associated with the inhibition of translation initiation. Polysome profiling also showed a small increase of heavy polysomes after 30 min of GC7 treatment, indicative of either ribosome slowing or stalling on the mRNA (Fig. 1i). Conversely, an increase in free 80S ribosomes was not observed until 6 h, correlating with p-eIF2α and ATF4 levels, and suggesting that the inhibition of elongation occurs before the inhibition of initiation (Fig. 1i). To validate the presence of slow-moving ribosomes in the early stages of eIF5A inhibition, a harringtonine pulse-chase ribosome run-off assay was conducted at 3 h of GC7 treatment. In this assay, cells are treated with harringtonine to prevent 80S ribosome loading and ribosome elongation rates can be inferred from the subsequent loss of polysomes. Consistent with the data above, eIF5A inhibition resulted in a significant increase in ribosomes remaining on the mRNA after harringtonine treatment compared to untreated cells, suggesting that treatment with GC7 slowed elongation rates and ribosome run-off (Fig. 1j and Fig. S2a, b).

To determine if the effects of GC7 were specific to the inhibition of hypusination, we first used siRNAs to deplete eIF5A1 and 2. Prolonged eIF5A1/2 depletion recapitulated the effects of GC7 by increasing p-eIF2α and ATF4 expression, as well as decreasing overall rates of protein synthesis (Supplementary Fig. 3a–f). Furthermore, depletion of the enzymes responsible for hypusination (DHPS and DOHH) also signals to increase p-eIF2α, suggesting that the effects observed with GC7 are due to the specific impact on hypusination (Supplementary Fig. 3g, h).

Finally, to evaluate whether eIF5A inhibition leads to ribosome stalling and activates the ribosome quality control system (RQC), we employed our established ribosome stalling assay[29] in HeLa cells. As HeLa cells displayed a dose-dependent inhibition of cell proliferation up to 40 μM GC7 (Supplementary Fig. 4a), we reasoned that this higher concentration would induce the greatest number of stalled ribosomes and therefore used this dose for the stalling assays. Consistent with the response in A549 cells, HeLa cells exhibited a similar reduction of hypusination and increase in ATF4 expression following GC7 treatment (Supplementary Fig. 4b–e). Whereas anisomycin robustly

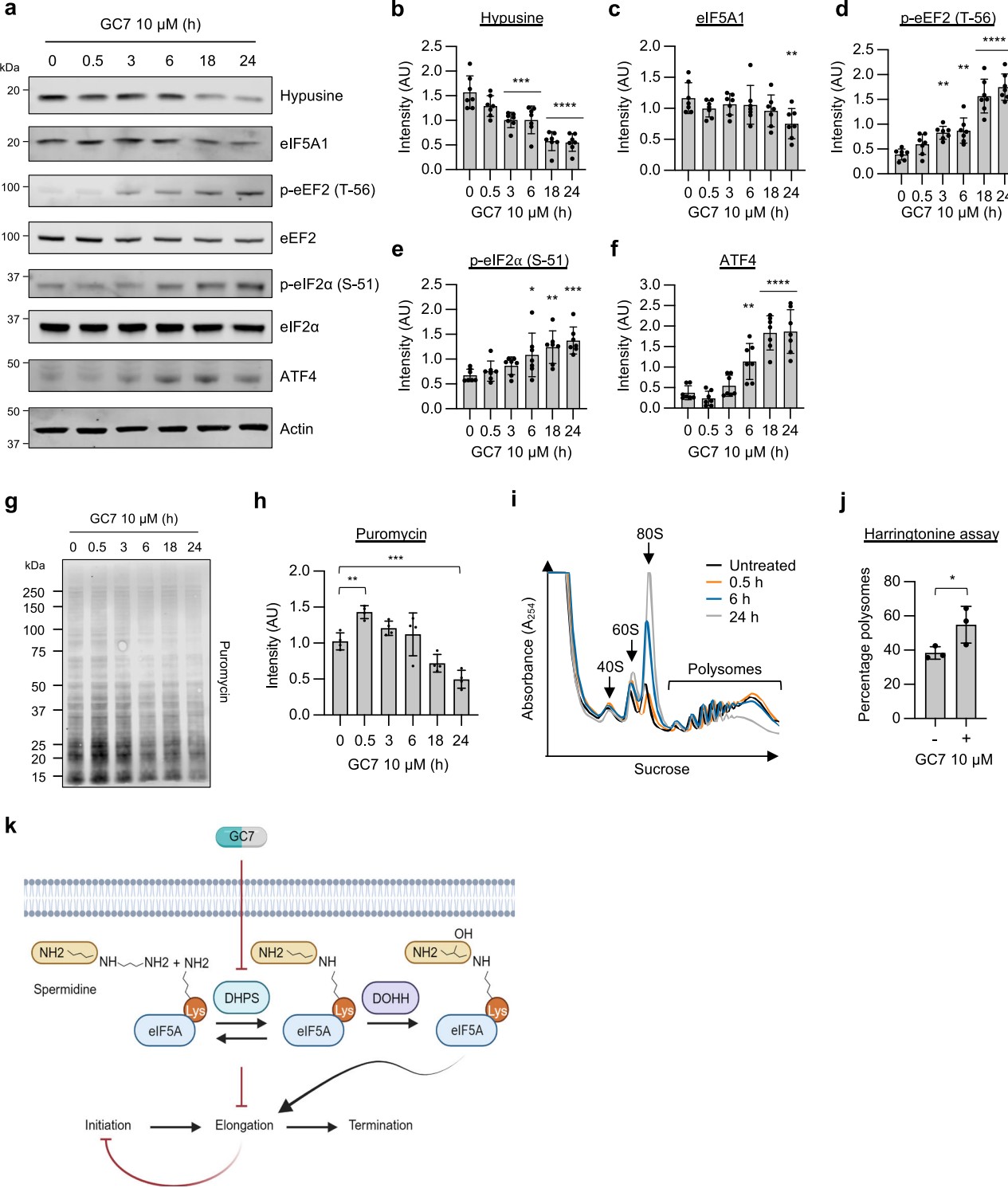

**Fig. 1 | Hypusination inhibition leads to sequential elongation and initiation blocks in A549 cells. a** Representative western blots for the indicated targets in A549 cells treated with 10 μM GC7 for the indicated time points. **b**–**f** Densitometry from the indicated targets presented in (**a**). Error bars represent means ± SD (*n* = 7 independent experiments) and are plotted with individual values. Statistical analysis was carried out using one-way ANOVA with Dunnett's multiple comparisons test (* = *p* < 0.05, ** = *p* < 0.01, *** = *p* < 0.001, **** = *p* < 0.0001) relative to the untreated (0 h) sample. **g** Representative western blot analysis of puromycin incorporation in A549 cells treated with 10 μM GC7 for the indicated time points. **h** Quantification of puromycin intensity presented in (**g**) normalized to total protein levels. Error bars represent means ± SD (*n* = 4 independent experiments) and are plotted with individual values. Statistical analysis was carried out using one-way ANOVA with Dunnett's multiple comparisons test (** = *p* < 0.01, *** = *p* < 0.001, **** = *p* < 0.0001) relative to the untreated sample. **i** Polysome profiling of A549 cells either untreated or treated with 10 μM GC7 for 0.5, 6 or 24 h. **j** Percentage of polysomes remaining in A549 cells following 5-min treatment with Harringtonine in either untreated cells or cells pre-treated with 10 μM GC7 for 3 h. Error bars represent means ± SD (*n* = 3 independent experiments) and are plotted with individual values. Statistical analysis was carried out using one-tailed unpaired student's *t* test (* = *p* < 0.05). **k** Schematic representation of hypusination inhibition and its downstream effects on translation. Source data are provided within the Source Data file.

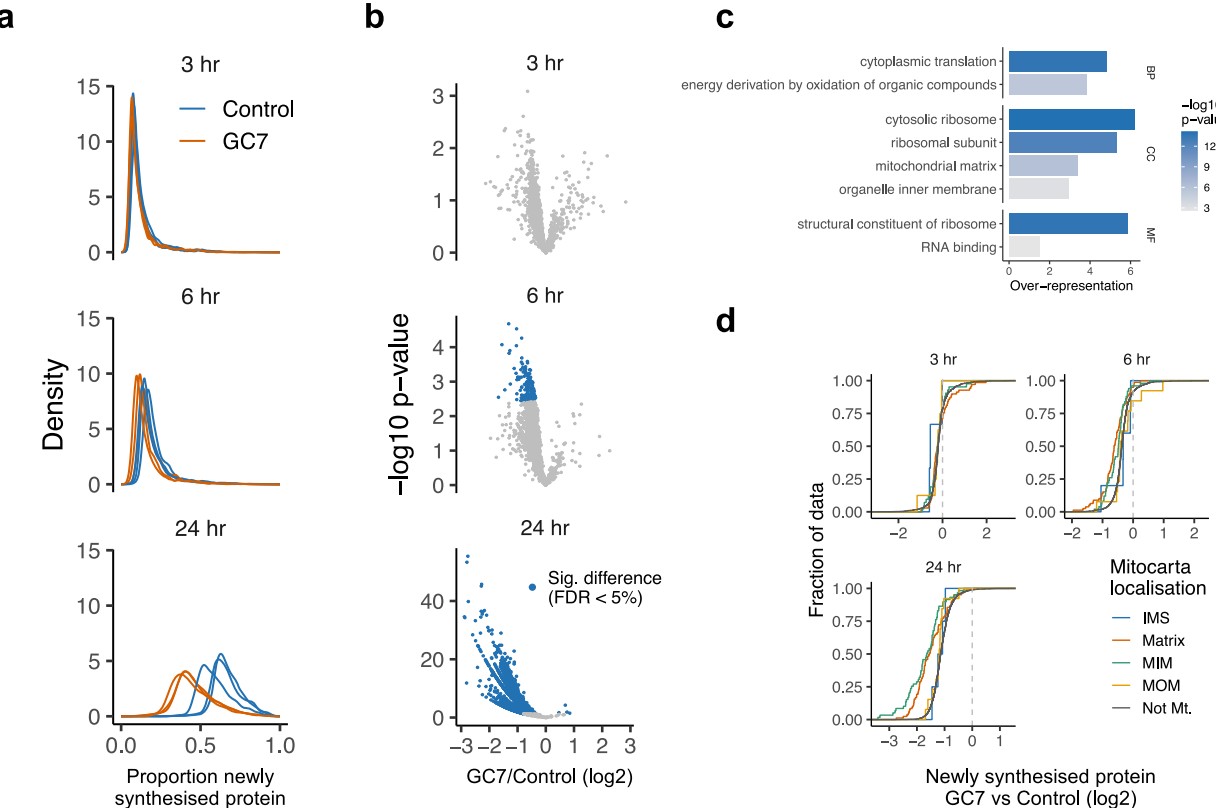

**Fig. 2 | Hypusination inhibition causes significant loss of newly synthesized proteins from 6 h. a** The proportion of newly synthesized proteins in GC7 and control treated A549 cells determined by dynamic SILAC. Three independent biological replates were used for each condition and each time point. **b** Volcano plots showing proteins significantly changing after GC7 treatment at each time point. Differential abundance of newly synthesized proteins was analyzed using DEqMS with the $\log_2$-transformed ratio of heavy to light intensities. Tests were two-sided and p-values were adjusted for multiple testing using the Benjamini–Hochberg False Discovery Rate (FDR) procedure. **c** Gene Ontology (GO) terms significantly over-represented among proteins showing a significant loss of synthesis at 6 h. Over-representation analysis was performed using goseq and the reported one-sided p-values were adjusted for multiple testing using the FDR procedure. BP Biological Process; CC Cellular Compartment; MF Molecular Function. **d** Empirical cumulative density plots for the difference in newly synthesised protein between GC7 and control for proteins with known mitochondrial sub-cellular localizations.

induced ribosome stalling (Supplementary Fig. 4f), GC7 treatment resulted in only a marginal increase in disomes due to ribosome collisions (Supplementary Fig. 4g). Furthermore, no phosphorylation of p38[MAPK] (Supplementary Fig. 4e), a marker of ZAK activation following ribosome collisions, was detected at earlier timepoints. Overall, these data suggest that eIF5A inhibition results in a small increase in ribosome collisions, which is insufficient to activate ZAK.

Taken together, these data suggest that inhibition of eIF5A activity slows ribosome elongation and translation initiation is subsequently inhibited, possibly in an eIF2α-dependent manner (Fig. 1k).

## eIF5A inhibition results in ribosomal and mitochondrial protein downregulation without evidence for eIF5A-dependent Di or tripeptide sequences

To explore the impact of eIF5A inhibition on translation, dynamic SILAC was used to quantify the newly synthesised proteome at 3, 6 and 24 h. This assay requires cells to be pulsed with heavy labelled SILAC media for the duration of GC7 treatment, to determine the change in heavy labelled peptides and hence infer changes to the total proteome. A PCA comparison shows the consistency of biological replicates and suggests most variation is associated with PC1, which separates the time points of treatment (Supplementary Fig. 5a). There was a significant change in the proportion of newly synthesised proteins after 6 h of GC7 (Fig. 2a), by which time 182/1978 quantified proteins showed a significant decrease in the proportion produced (Fig. 2b). At 24 h, 1950/2121 proteins showed a significant decrease in the proportion of newly synthesised proteins, indicating that translation was

globally inhibited, in agreement with the phosphorylation of eIF2α, and eEF2 (Fig. 1a, d, e). GO term over-representation analysis was used to identify functional groups enriched in the proteins significantly impacted by 6 h GC7 treatment. This indicated that cytosolic ribosomal proteins and mitochondrial matrix proteins were the most significantly impacted by GC7 treatment (Fig. 2c). Since mitochondrial matrix proteins were particularly affected, we separated mitochondrial proteins by their sub-mitochondrial localisation and observed that matrix and inner-membrane proteins were significantly more impacted by GC7 than inner-membrane space and outer membrane proteins at 6 h and 24 h ($p < 0.05$; Two-sample Kolmogorov-Smirnov test; Fig. 2d).

Previously, eIF5A has been implicated in the efficient translation of specific dipeptides and tripeptides, especially those containing proline[16,18]. However, we did not find an enrichment or any relationship between polyprolines and the effect of eIF5A inhibition[18] (Supplementary Fig. 5b, c). We therefore performed unsupervised statistical learning using ridge regression to identify the amino acid compositions that were predictive of a significant impact of eIF5A inhibition at 6 h. We used the frequency of each amino acid and each possible dipeptide and the presence of each possible tripeptide as features and an 80:20 split between training and hold-out test data. We achieved an area under the curve for the Receiver Operator Curve of 0.64 against the test data, indicating that the model was able to predict the impact of eIF5A inhibition with better than random accuracy (Supplementary Fig. 5d). The positively charged amino acids lysine and arginine were amongst the best predictors (Supplementary Fig. 5e). However, when

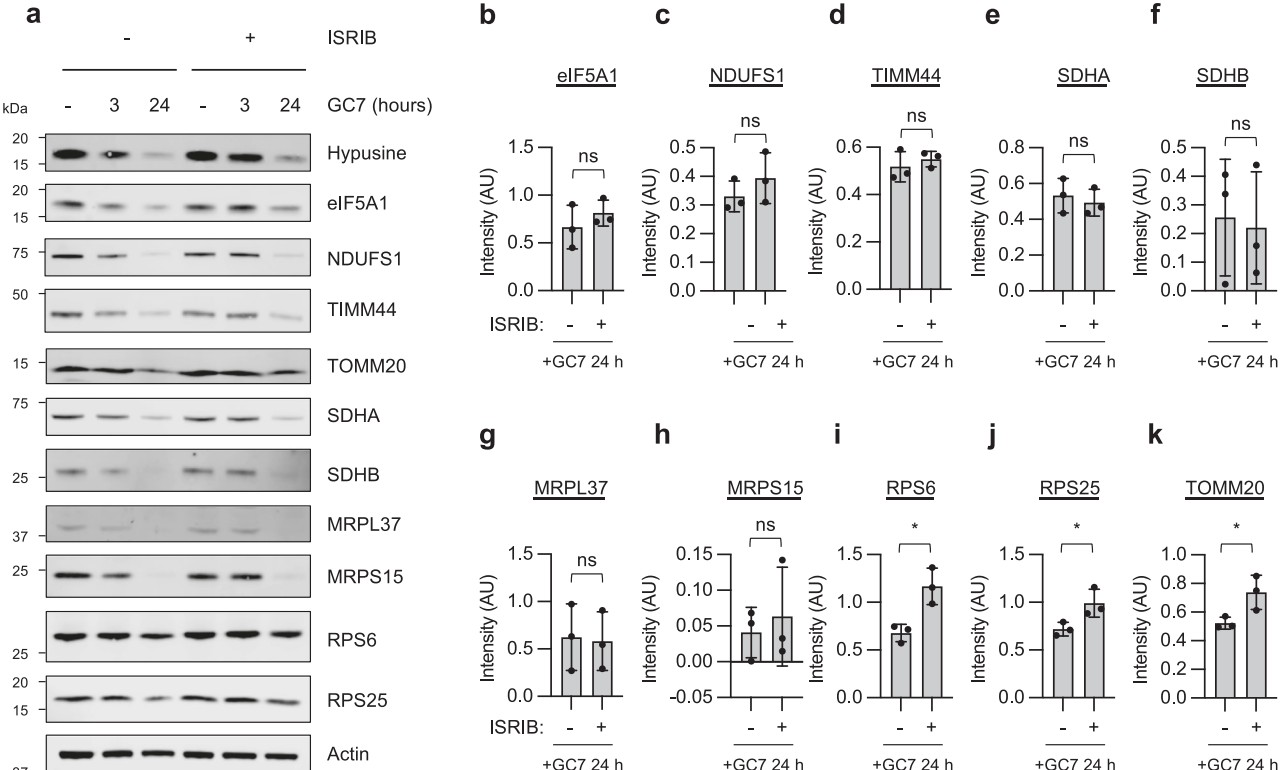

**Fig. 3 | ISRIB leads to recovery of p-eIF2α specific targets following inhibition of hypusination. a** Representative western blots of A549 cells treated with 10 μM GC7 for either 3 h or 24 h in the presence of ISRIB (200 nM) or normal growing conditions. **b–k** Densitometry of 24-h GC7 treatments (+/- ISRIB) from (**a**) for the indicated proteins. Error bars represent means ± SD ($n = 3$ independent experiments) and are plotted with individual values. Statistical analysis was carried out using a two-tailed unpaired t-test (ns = not significant, * = $p < 0.05$). Source data are provided within the Source Data file.

the proteins were divided into functional groups based on the enriched GO terms (Fig. 2c), there was no relationship between the frequency of lysine and arginine and the impact of eIF5A inhibition (Supplementary Fig. 5f, g). Furthermore, we did not observe the relationship previously implicated between amino acid composition and the impact of eIF5A inhibition or knockdown on tripeptides and amino acids at any time point (Supplementary Fig. 5h). These data strongly suggest our regression merely identified weak associations between amino acid composition and functional groups of affected proteins. In summary, while the impact of eIF5A inhibition on the newly synthesized proteome is clear, there is no strong evidence that this is due to a dependence of specific amino acid motifs on eIF5A for efficient translation.

**Uncoupling translation initiation inhibition from eIF5A inhibition revealed eIF2α-dependent mRNAs**

The integrated stress response (ISR) is a conserved signalling pathway that allows cellular adaptation following exposure to a diverse range of stress stimuli. The ISR centres on the phosphorylation of eIF2α, which inhibits general protein synthesis while enabling the selective translation of mRNAs encoding stress-response proteins to restore homeostasis[30]. Four eIF2 kinases (HRI, PKR, PERK and GCN2) are activated in response to a diverse range of stress stimuli including oxidative stress, viral infection, unfolded proteins, amino acid deprivation and ribosome collisions[31]. Given that we show there is feedback inhibition from a reduction in eIF5A activity to p-eIF2α (Fig. 1), we reasoned that the absence of sequence specificity among the targets identified could reflect that the dual impact of initiation and elongation inhibition was confounding our data. Therefore, we used the integrated stress response inhibitor (ISRIB), which antagonizes the impact of ISR activation[32], in combination with GC7 treatment to uncouple these

events (Fig. 3). As the inhibition of translation initiation occurred within 6 h of GC7 treatment (Fig. 1e), A549 cells were pre-treated with ISRIB for 2 h to ensure complete alleviation of ISR signalling prior to the addition of GC7 (Fig. 3a). Since mitochondrial protein expression has been shown previously to be impacted by eIF5A inhibition[22], we tested a range of mitochondrial proteins that were shown to be either downregulated (NDUFS1, TIMM44 and SDHA, MRPS15) or absent (SDHB, MRPL37 and TOMM20) from our SILAC dataset (Supplementary Data 1). In addition, we tested two cytosolic ribosomal proteins (RPS6 and RPS25) that were also downregulated after 24 h GC7 treatment. We observed a reduction in hypusination and the protein levels of all downregulated targets from our SILAC data at 24 h (Fig. 3a and Figure S6a). Ribosomal protein levels and TOMM20, which exhibited a more modest reduction in the SILAC experiment, also showed a small decrease at 24 h (Fig. 3a). We hypothesized that the addition of ISRIB would alleviate the effects of eIF2α phosphorylation and subsequent inhibition of translation initiation. We identified 7 proteins (eIF5A, NDUFS1, TIMM44, SDHA, SDHB, MRPL37, and MRPS15) downregulated after GC7 treatments that failed to recover with ISRIB, indicating the reduction in expression occurs via an eIF2B independent mechanism (Fig. 3a–h). Conversely, downregulation of RPS6, RPS25, and TOMM20 was significantly restored in the presence of ISRIB, suggesting these proteins are eIF2α-dependent and their downregulation is consistent with global translation inhibition (Fig. 3i–k). In addition, treatment with a different inhibitor of hypusination, ciclopirox (CPX), also increases p-eIF2α and ATF4 expression, decreases expression of mitochondrial proteins, and inhibits protein synthesis (Supplementary Fig. 6b–k). These findings imply that eIF5A inhibition results in the downregulation of both eIF2α-dependent and eIF2α−independent targets, providing an explanation for why bulk proteome analysis following eIF5A inhibition may not reveal specific eIF5A sequences in the affected proteins.

## The reduction in translation initiation following eIF5A inhibition is driven by PERK- and HRI-mediated eIF2α phosphorylation

Following the delineation of eIF5A-specific and non-specific effects on protein synthesis, we aimed to elucidate the mechanism for feedback inhibition from elongation to initiation. As our previous data suggested the inhibition of translation was most likely dependent on the phosphorylation of eIF2α (Fig. 1), all four eIF2 kinases (HRI, PKR, PERK and GCN2) were depleted using siRNAs. We then assessed p-eIF2α levels and the subsequent induction of ATF4 expression following GC7 treatment. These data show that knockdown of HRI using two different siRNAs (Fig. 4a, b) reduced the induction of p-eIF2α by GC7, although only one siRNA showed a significant reduction (Fig. 4c). Moreover, HRI depletion abolished GC7-induced ATF4 expression (Fig. 4d). In addition, knockdown of PERK (Fig. 4e, f) also reduced the induction of p-eIF2α by GC7 to untreated levels (Fig. 4g) and significantly reduced the induction of ATF4 (Fig. 4h). Conversely, knockdown of either PKR (Supplementary Fig. 7a–g) or GCN2 (Supplementary Fig. 7e–h) failed to reduce GC7-induced p-eIF2α and ATF4 expression. Taken together, these data suggest that both HRI and PERK are activated in response to GC7 treatment and elongation inhibition, and may be responsible for mediating the inhibition of translation initiation.

## Mitochondrial dysfunction results in activation of OMA1 and DELE1 proteases that activate HRI and induce ATF4 expression

While inhibition of elongation and initiation activate the unfolded protein response (UPR), leading to PERK activation, eIF2α phosphorylation, and ATF4 expression[30], the mechanism of HRI activation in our system remains unclear and was investigated further. Previous studies have linked eIF5A inhibition to mitochondrial dysfunction[22,33]. Therefore, mitochondrial morphology was examined using immunofluorescence confocal microscopy targeting TIMM44, which revealed that after 24 h of GC7 treatment that the majority of cells exhibited large, fragmented mitochondria (Fig. 5a, b). GC7 treatment did not result in the generation of ROS, whereas the redox cycling agent menadione stimulated extensive ROS production (Fig. 5c). These data suggest that GC7 causes mitochondrial dysfunction that is independent of ROS accumulation.

A previous study has shown that ROS-independent mitochondrial dysfunction is associated with the activation of the mitochondrial proteases OMA1 and DELE1[28]. Upon activation, OMA1 cleaves DELE1, which translocates to the cytoplasm and activates HRI[28]. To investigate whether this signalling cascade is activated by eIF5A inhibition, we utilized siRNAs to deplete both OMA1 and DELE1, confirming their down-regulation through qPCR (Supplementary Fig. 8a, b). OMA1 protein depletion was confirmed by western blotting (Supplementary Fig. 8c), however, we were not able to determine DELE-1 protein depletion due the lack of specific commercially available DELE-1 antibodies[28]. However, as DELE-1 protein has been suggested to be degraded within 30 min[29,34], we reasoned analysis of mRNA levels should be sufficient. Subsequently, cells were treated with GC7 for 24 h, and ATF4 induction was assessed via western blot. Following eIF5A inhibition, ATF4 protein levels increased approximately 5-fold in cells treated with a control siRNA and GC7 (Fig. 5d–i). However, ATF4 induction was significantly reduced upon OMA1 and DELE1 knockdown (Fig. 5d–i). Collectively, these findings support the notion that OMA1 and DELE1 are activated in response to mitochondrial dysfunction, contributing to HRI activation and ATF4 expression following GC7 treatment (Fig. 5j).

## EIF5A inhibition induces cell cycle arrest through mitochondrial dysfunction and eIF2α phosphorylation

Our findings suggest that cancer cells with high OXPHOS levels or a reliance on mitochondria for energy production would exhibit increased sensitivity to eIF5A inhibition. To test this hypothesis, we examined the response of A549 derived from lung adenocarcinoma, which is GC7-sensitive, to a cell line derived from healthy tissue, MCF10A. The xCELLigence RTCA platform quantifies cell proliferation in real time, which is expressed in arbitrary units termed cell index. We compared proliferation of both A549 and MCF10A cells following treatment with GC7 and observed a reduction of cell index in A549 cells, whereas MCF10A were less sensitive (Fig. 6a–c). The reduction of cell index in A549 at 48 h was not a result of cell death (Supplementary Fig. 9a), but rather a decrease of cell proliferation, as indicated by an accumulation of cells in the G1 phase of the cell cycle over a 72 h period (Supplementary Fig. 9b). Conversely, the effect of GC7 on MCF10A was more modest. Although there was no impact on cell cycle distribution (Fig. 9c), there was a small decrease in the rate of proliferation (Fig. 6b), in line with the observed inhibition of protein synthesis (Supplementary Fig. 9d–l).

Cells are reliant on mitochondria for ATP production when they are cultured in galactose-containing media which can highlight mitochondrial liabilities[35]. Given that GC7 inhibits mitochondrial protein expression of A549 cells (Fig. 2), we compared ATP production in A549 and MCF10A cells grown in either glucose or galactose and treated with GC7. Rotenone was used as a positive control for OXPHOS inhibition. Importantly, ATP production is not impaired by GC7 in A549 cells grown in galactose at early timepoints, strongly suggesting GC7 is not a direct ETC inhibitor (Supplementary Fig. 10a). However, ATP production is impaired in galactose from 6 h after GC7 treatment, but not in cells grown in glucose media, suggesting mitochondrial dysfunction following GC7 treatment (Supplementary Fig. 10a) and correlating with previously observed changes in mitochondrial protein expression (Fig. 2). Conversely, MCF10A showed a small reduction in ATP production from 18 h treatment in both glucose and galactose containing media (Supplementary Fig. 10b). To determine the mitochondrial function of A549 and MCF10A cells we measured their oxygen consumption rates (OCRs). Although basal OCR was higher in A549 cells, MCF10A cells had a large reserve capacity of mitochondrial respiration, whereas A549 cells had no reserve capacity, suggesting that A549 cells may be very dependent on mitochondria for energy production as their mitochondria are already working at maximum capacity (Fig. 6d). To understand the impact GC7 had on the mitochondrial function we measured OCR following 6 h and 24 h GC7 treatment. GC7 treatment reduced the basal OCR of A549 cells within 6 h, correlating with the reduction of mitochondrial proteins in our SILAC analysis (Fig. 6e and Figure S10c). Conversely, MCF10A cells were minimally affected (Fig. 6e and Figure S10d), suggesting they rely less on their mitochondria in the steady state. As there was no decrease in ATP production of A549 cells treated with GC7 in glucose media (Supplementary Fig. 10a), but mitochondrial function is inhibited (Fig. 6e), we assessed their ability to utilise glycolysis by determining the extracellular acidification rate (ECAR). A549 cells switched to glycolysis within 6 h of GC7 treatment, whereas MCF10A did not enhance glycolysis until 24 h, perhaps due to their reduced basal respiration requirements (Supplementary Fig. 10e–g). These data further support the hypothesis that cancer cell lines that are dependent on their mitochondria in the steady state, would be more sensitive to GC7. Our data suggests that cell lines with a dependence on mitochondria, such as MpM[36], or those with minimal reserve capacity, such as HeLa[37], were sensitive to GC7 (Supplementary Figs. 4a, 11a–d). Conversely, cell lines with larger reserve mitochondrial capacities, such as MCF7[38] and HT-29[39], were insensitive to GC7 (Supplementary Fig. 4a). Additionally, tissue microarrays containing samples derived from lung adenocarcinoma showed a correlation between eIF5A1 and Ki67 positive cells (Supplementary Fig. 11e, f and Supplementary Data 2) and SDHB (Supplementary Fig. 12 and Supplementary Data 3), suggesting potential sensitivity in this context.

Our data suggest that A549 cells are reliant on their mitochondria for energy production and that GC7-induced mitochondrial dysregulation may limit cancer cell growth. To investigate if the inhibition of cell proliferation in A549 cells was eIF2-dependent[32,40–42], cells were

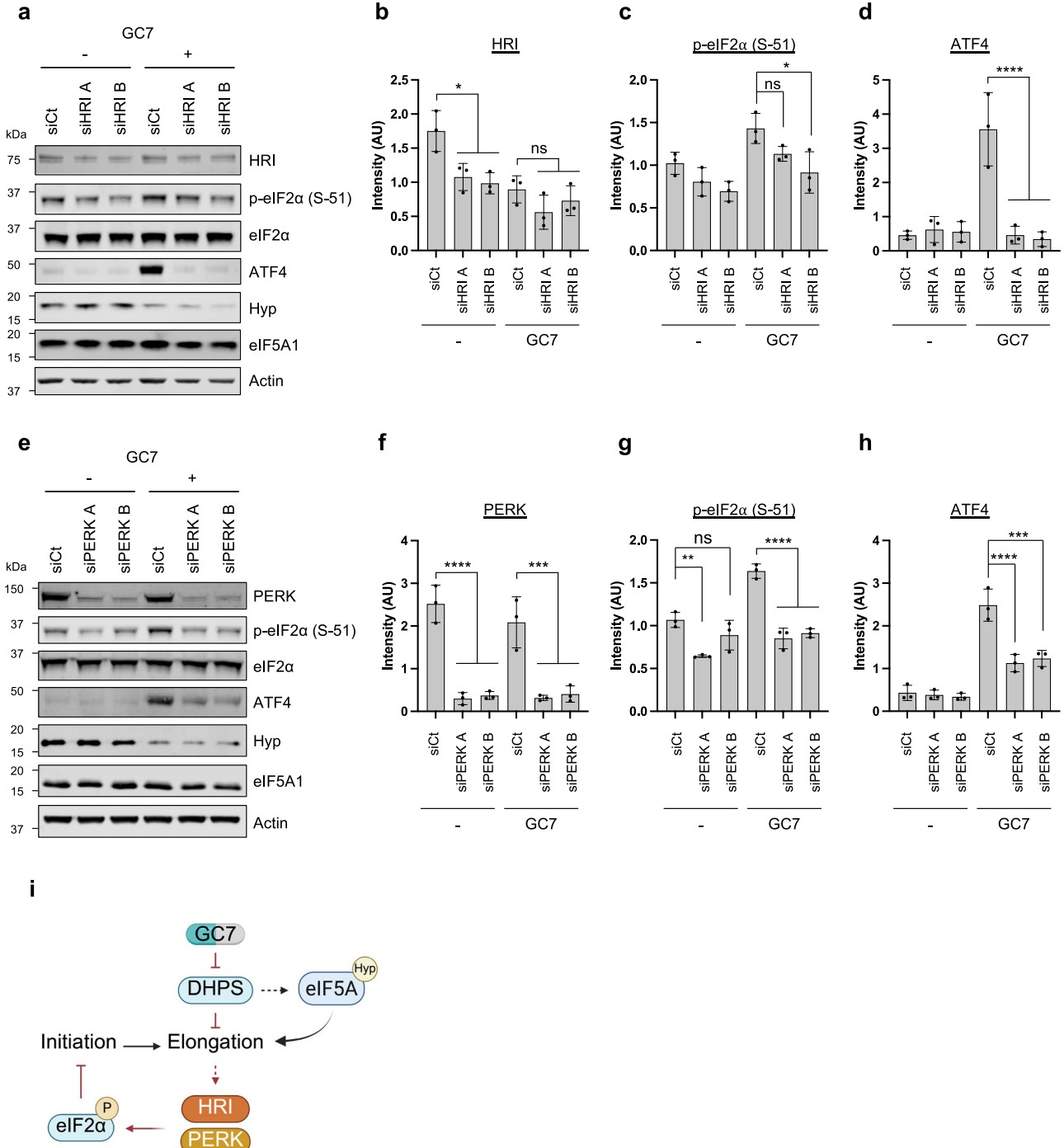

**Fig. 4 | Translation initiation inhibition following eIF5A inhibition is driven by PERK and HRI activation and eIF2α phosphorylation. a** Representative western blots for the indicated targets from A549 cells transfected with siRNAs against HRI and treated with 10 μM of GC7 for 24 h. **b**–**d** Densitometry of western blots for the indicated targets from (**a**). **e** Representative western blots for the indicated targets from A549 cells transfected with siRNAs against PERK and treated with 10 μM of GC7 for 24 h. **f**–**h** Densitometry of western blots for the indicated targets from (**e**).

All error bars represent means ± SD ($n = 3$ independent experiments) and are plotted with individual values Statistical analysis was carried out using One-Way ANOVA with Tukey's multiple comparisons test (ns = not significant, * = $p < 0.05$, ** = $p < 0.01$, *** = $p < 0.001$, **** = $p < 0.0001$). **i** Schematic representation of the proposed regulation of eIF2 following treatment with GC7. Created in BioRender. Harvey, R. (2025) https://BioRender.com/44scex7. Source data are provided within the Source Data file.

co-treated with GC7 and ISRIB. The presence of ISRIB significantly reversed the inhibition of cell proliferation (Fig. 6f) and alleviated the accumulation of cells is the G1 phase of the cell cycle (Fig. 6g), suggesting the impact of GC7 on cell proliferation results from feedback inhibition of initiation, rather than a direct effect on elongation. To confirm the relationship between GC7-dependent phosphorylation of eIF2α and the inhibition of cell proliferation, we again utilised eIF2

S51A MEF cells. In support of our observations with ISRIB, GC7-induced inhibition of proliferation was significantly reduced in the S51A cell line (Fig. 6h), again suggesting that the effect on cell proliferation is driven by the phosphorylation of eIF2α. To confirm the role of the eIF2α kinases in this mechanism, siRNAs for all four eIF2α kinases were again utilised. Consistent with previous data (Fig. 4), silencing either HRI or PERK rescued GC7-induced inhibition of proliferation (Fig. 6i, j),

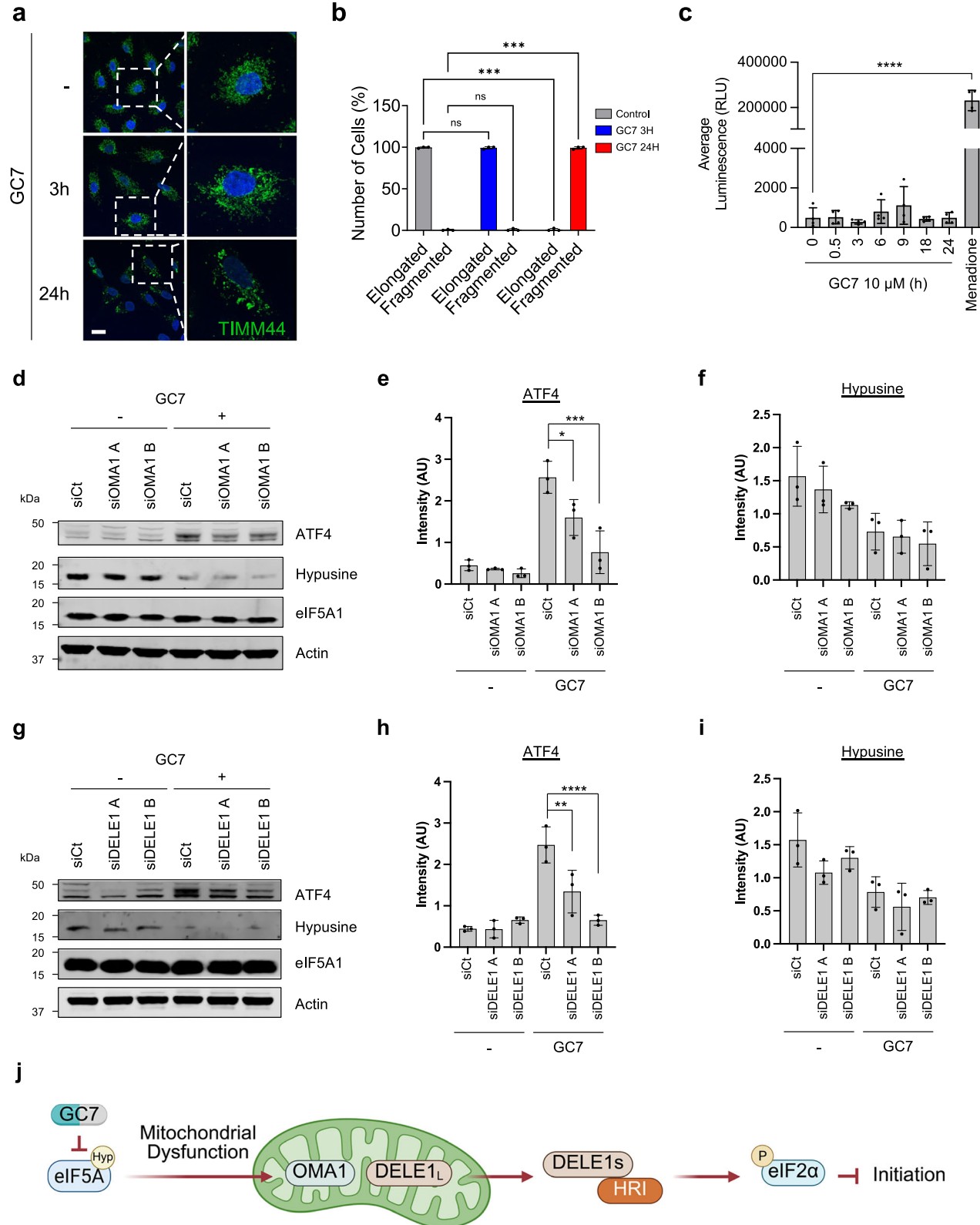

whereas silencing PKR and GCN2 had no effect on cell proliferation (Supplementary Fig. 10h–i). Finally, to confirm that mitochondrial dysfunction and the activation of HRI via the OMA1 and DELE1 cascade was responsible for the observed effects on cell proliferation, we used siRNAs specific to OMA1 and DELE1. As expected, depletion of both OMA1 and DELE1 reversed GC7-induced inhibition of proliferation (Fig. 6k, l).

These data collectively suggest that following GC7 treatment, activation of OMA1 and DELE1 in response to mitochondrial dysfunction in turn activates HRI to inhibit cancer cell proliferation.

## Discussion

Cancer progression is associated with aberrant control of mRNA translation and proteins that regulate this process present promising

**Fig. 5 | Mitochondrial dysfunction leads to OMA1 and DELE1 dependent HRI activation. a** Confocal microscopy of A549 cells stained with TIMM44 (Green) and treated with 10 μM of GC7 for 3 and 24 h. Nuclei were stained with Hoechst (Blue). Scale bar: 20 μm. **b** Quantification of mitochondria shape from (**a**). Error bars represent means ± SD ($n = 3$ independent experiments) and are plotted with individual values. Statistical analysis was carried out using two-way ANOVA with Sidak's multiple comparisons test (ns = not significant and *** = $p < 0.001$). **c** ROS detection assay in A549 cells treated with 10 μM GC7 for the indicated time points. Error bars represent means ± SD ($n = 3$ independent experiments) and are plotted with individual values. Statistical analysis was carried out using one-way ANOVA with Dunnett's multiple comparisons test (**** = $p < 0.0001$). A 2-h treatment with 20 μM menadione was used as a positive control for ROS production. **d** Representative western blots for the indicated targets from A549 cells transfected with siRNAs against OMA1 and treated with 10 μM of GC7 for 24 h. **e, f** Densitometry of western blots for the indicated targets from (d). Error bars represent means ± SD ($n = 3$ independent experiments). **g** Representative western blots for the indicated targets from A549 cells transfected with siRNAs against DELE1 and treated with 10 μM of GC7 for 24 h. **h, i** Densitometry of western blots for the indicated targets from (**g**). Error bars represent means ± SD ($n = 3$ independent experiments) and are plotted with individual values. Statistical analysis (in **e–i**) was carried out using one-way ANOVA with Tukey's multiple comparisons test (* = $p < 0.05$, ** = $p < 0.01$, *** = $p < 0.001$, **** = $p < 0.0001$). **j** Schematic representation of OMA1 and DELE1 function. Created in BioRender. Harvey, R. (2025) https://BioRender.com/r1ph8ig. Source data are provided within the Source Data file.

therapeutic avenues for effective anti-cancer strategies[1,2,43,44]. While existing interventions that target mRNA translation predominantly concentrate on targets associated with the initiation phase, emerging research indicates that disruptions in elongation regulation would also provide viable treatment options[1,2,43,44]. In this study we explored the impact of eIF5A inhibition on overall translation and its potential ramifications for cancer cell growth.

Although GC7 has previously been proposed to inhibit translation initiation[45], we have identified the mechanistic link between the reduction in eIF5A activity, and mRNA translation elongation, with feedback to an inhibition of translation initiation (Figs. 1, 7). Specifically, upon eIF5A inhibition there is a progressive shift from a reduction in the elongation phase of translation to a more generalized inhibition of protein synthesis through eIF2α phosphorylation by activation of HRI and PERK (Figs. 4, 5). We did not investigate the role of FAM69C[46] and MARK2[47], both recently proposed to function as eIF2α kinases, however they should be considered in future investigations. Although we also observed the dephosphorylation of 4EBP1 after 18 h GC7 treatment, it was unaffected at 6 h, suggesting the initial GC7-induced inhibition of translation initiation may be independent of mTOR. However, considering the central role of mTOR in the regulation of eEF2K and mitochondria function[2,48], the role of mTOR is currently the subject of further investigation.

Although it has been shown in both yeast and mammalian systems that eIF5A is required to suppress pausing caused by the presence of proline rich motifs[16–18,49], we find no evidence that the loss of eIF5A activity affects the synthesis of proteins that harbour such motifs (Fig. 2). However, the dynamic SILAC approach will predominantly detect highly abundant proteins and may fail to identify poly-proline proteins that are poorly expressed or secreted, so the entire proteome has not been sampled and is not represented in these data. Taken together these data suggest that the ribosomal pause sites which occur in mammalian cells depleted of eIF5A[49], have little impact on the synthesis or accumulation of the corresponding proteins. Our data are also consistent with a proteomics analysis following prolonged knock down of eIF5A in HeLa cells which similarly did not find a selective downregulation of proteins containing poly-proline stretches[50].

Our data confirm previously reported eIF5A-downstream targets[22,24,51] and we additionally show that there is a correlation with a reduction in eIF5A activity and the synthesis of ribosomal, inner mitochondrial membrane, and matrix proteins, which also do not feature proline di- or tripeptides (Fig. 2 and Figure S5). We postulated that their decreased synthesis may not be directly linked to eIF5A functionality and a decrease in elongation but rather arise from inhibition of translation initiation (Figs. 1, 7), and we used ISRIB to ameliorate the impact of inhibition of translation initiation (Fig. 3). Specifically, we show that the levels of RPS6, RPS25, and the mitochondrial protein TOMM20 are restored following ISRIB treatment implying that these proteins are indirect targets of eIF5A inhibition, with their diminished expression attributable to the global inhibition of protein synthesis (Fig. 3a, l–k). As ribosomal proteins are generally

stable, they can have a relatively long half-life which has the potential to mask the effects of the synthesis of these proteins[52]. As such, it is important to determine newly synthesised proteins by either the proteomics approach utilised in this study, or translation efficiencies by ribosome profiling. Importantly, we also identified proteins that remained downregulated in the presence of GC7 after ISRIB addition, suggesting either direct regulation by eIF5A or elongation, or potentially via an alternative pathway independent of translation initiation (Fig. 3). For proteins that exhibit no recovery upon ISRIB addition the regulatory mechanism by which eIF5A controls their expression remains unclear. However, the observed loss of many mitochondrial proteins (Fig. 2c, d), particularly inner mitochondrial and matrix proteins, as opposed to outer mitochondrial membrane proteins, suggests specificity to the proteins downregulated. Given that the affected proteins do not exhibit a higher frequency of specific di- or tripeptides (Fig. 2 and Figure S5), it is conceivable that the loss of these proteins may result from disruption of the TIM23 complex, responsible for translocating proteins across inner mitochondrial membranes and into the matrix. Indeed, we observed a decrease in the expression of some TIM23 complex members following GC7, including Tim44 and Tim50 (Supplementary Data 1). Previous studies have indicated that disruption of the inner mitochondrial membrane transport complex leads to the downregulation of several mitochondrial proteins[53]. Therefore, eIF5A inhibition could induce the downregulation of proteins within the TIM23 complex, subsequently resulting in the broader downregulation of inner mitochondrial proteins. Consistent with this, a recent study proposes that the loss of eIF5A inhibits Tim50 in yeast affects mitochondrial protein import[54], however, the prolines in human Tim50 are distributed throughout the sequence rather than clustering together as in yeast, so it is possible the mechanisms of regulation may differ.

The association between eIF5A inhibition and mitochondrial dysfunction, either as a primary or secondary effect, suggests the elongation factor could be a putative target for mitochondrial-dependent cancer therapeutic strategies. Our findings show that mitochondrial dysfunction is sensed through the OMA1-DELE1-HRI-eIF2α cascade, which limits cancer cell proliferation (Fig. 5d, g), and that cell lines reliant on OXPHOS for energy production, or those with minimal reserve mitochondrial capacity, are more sensitive to GC7 (Supplementary Figs. 4a, S11a−d). Although not an exhaustive comparison of cell lines, these data suggest that sensitivity to GC7 is related to mitochondrial function and provides the basis for future studies to explore the viability of targeting mitochondria-dependent cancers with inhibitors such as GC7.

As GC7 has been proposed to regulate cellular events independently of eIF5A[55], we used a 20-fold lower concentration of GC7 in our study. In addition, we utilised several independent approaches to confirm the specificity of GC7 in our model. We first utilised siRNA mediated knockdown of eIF5A1/2 or the hypusination enzymes (DHPS and DOHH), which all recapitulated the response to GC7. Furthermore, we utilised a distinct small-molecule inhibitor of

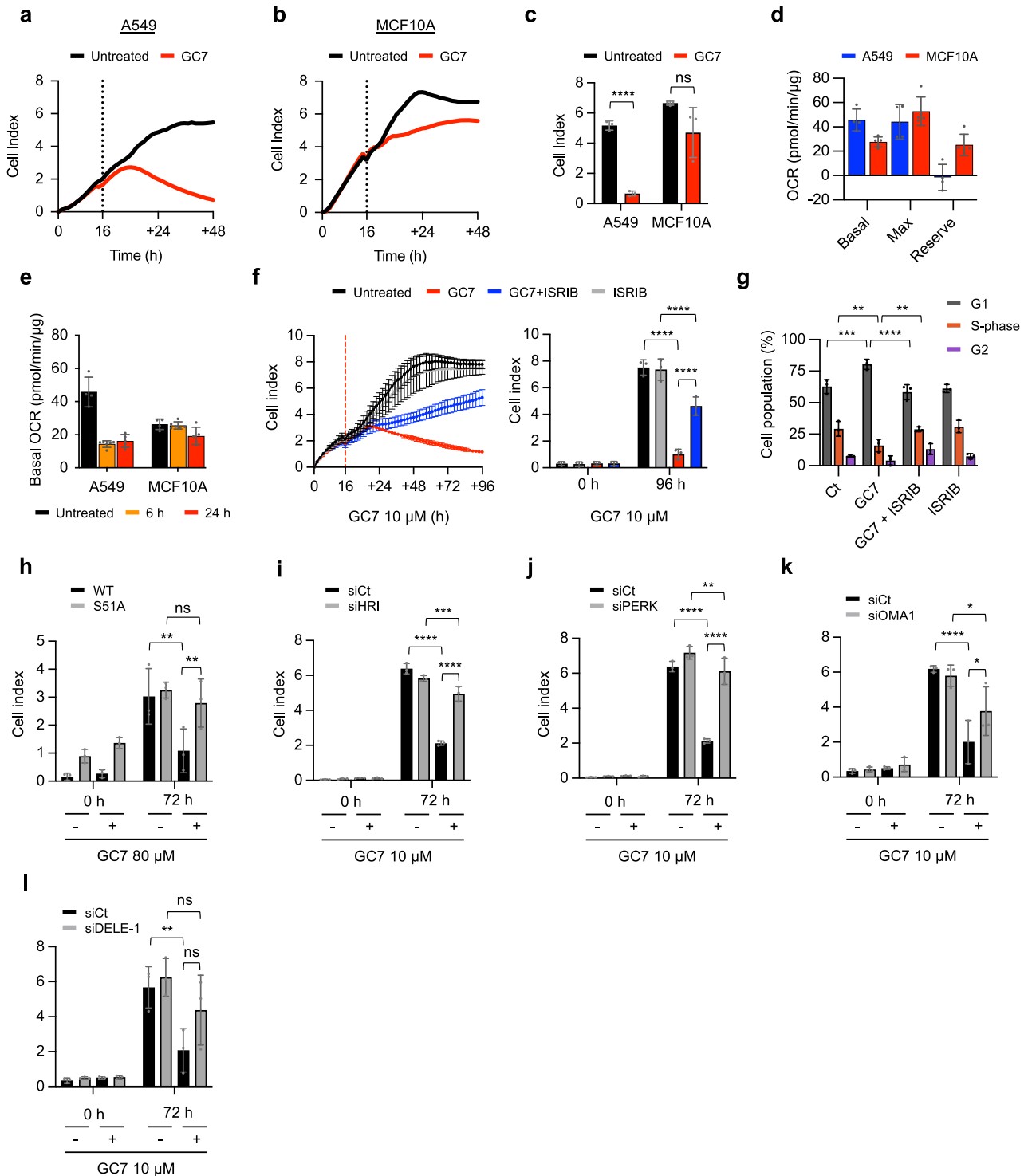

hypusination (CPX) targeting DOHH rather than DHPS (target of GC7), which also recapitulates observations with GC7. Taken together, these independent lines of approach strongly suggest that our observations are not due to indirect effects of GC7 and are instead due to its inhibition of eIF5A hypusination.

Overall, our data show that prolonged inhibition of the elongation phase of mRNA translation through a reduction in eIF5A activity impacts on initiation resulting in a shutdown of global protein synthesis (Fig. 7). This interconnected relationship underscores the importance of adopting a holistic approach when investigating the regulatory mechanisms of mRNA translation. A nuanced understanding of these relationships offers a more detailed insight, enabling

the development of targeted methodologies for the study of mRNA translation and the creation of therapies focused on specific stages of this process.

## Methods

### Ethics

The study design and conduct complied with all relevant regulations regarding the use of human study participants and received ethics committee approval. For tissue microarrays, human resected primary Lung adenocarcinoma was used under ethical approval for the use of surplus tissue in research (REC ref. 16:/WS/0207) and managed by the NHSGGC Biorepository and Pathology Tissue Resource. Tissues were

**Fig. 6 | Reduced cell proliferation following eIF5A inhibition is driven by mitochondrial dysfunction and eIF2α phosphorylation.** Representative xCEL-Ligence RTCA impendence measurements of (**a**) A549 and (**b**) MCF10A cells treated with 10 μM GC7 (indicated by the dotted line) and continuously monitored in technical duplicate. Cell index is an arbitrary value of impedance generated from proliferating cells. **c)** Cell index at 48 h 10 μM GC7 treatment in A549 and MCF10A cells. Error bars represent means ± SD ($n = 3$ independent experiments). Statistical analysis was carried out using two-tailed unpaired student's $t$ test (ns = not significant and **** = $p < 0.0001$). **d** Basal, maximum and reserve capacity of oxygen consumption rates (OCR) were measured in A549 ($n = 4$ technical replicates) and MCF10A ($n = 5$ technical replicates) cells. OCR was normalized to total protein and error bars represent means ± SD. **e** Basal OCR of A549 and MCF10A cells treated with 10 μM GC7 for 6 and 24 h. OCR was normalized to total protein. Data shown represent mean ± SD from technical replicates for untreated ($n = 4$), 6 h ($n = 6$) and 24 h ($n = 5$). **f** Left, representative xCELLigence RTCA impendence measurements of

A549 cells treated with 10 μM GC7 (red), 200 mM ISRIB (grey), GC7 plus ISRIB (blue), or untreated (black). Error bars represent mean ± SD ($n = 3$ technical replicates) and treatment point is indicated by red line. Right, cell index at 96 h. Error bars represent means ± SD ($n = 3$ independent experiments). **g** Cell cycle distribution of A549 cells treated with 10 μM GC7 and 200 mM ISRIB for 72 h. DNA was stained using FxCycle violet dye and quantified using the Dean-Jett-Fox model. Error bars represent means ± SD ($n = 3$ independent experiments). Cell index from xCELLigence RTCA instrument of (**h**) MEF WT and MEF S51A cells, or A549 following transfection with siRNAs specific to (**i**) HRI, (**j**) PERK, (**k**) OMA1 and (**l**) DELE-1, treated with GC7 for 72 h. Error bars represent means ± SD ($n = 3$ independent experiments). Statistical analysis (in **f–l**) was carried out using two-way ANOVA with Tukey's multiple comparisons test (ns = not significant, * = $p < 0.05$, ** = $p < 0.01$, *** = $p < 0.001$, **** = $p < 0.0001$). Source data are provided within the Source Data file.

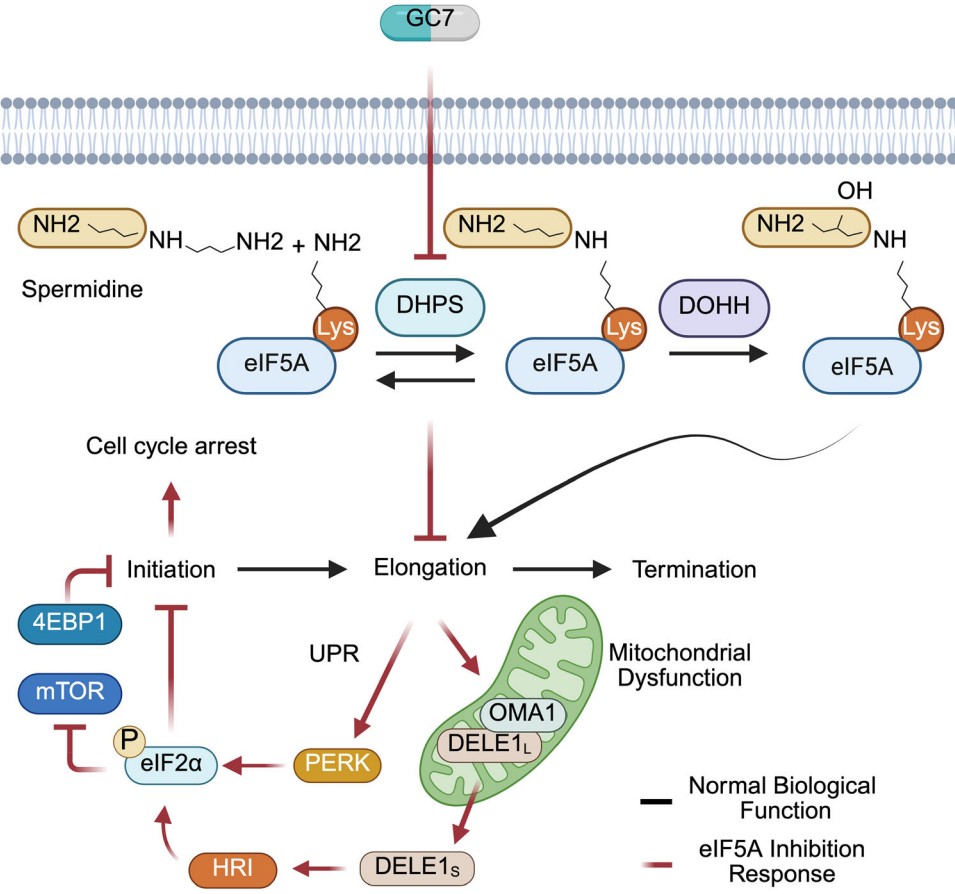

**Fig. 7 | A proposed model of the events following inhibition of hypusination and subsequent mitochondrial dysfunction.** Inhibiting eIF5A hypusination with GC7 leads to the inhibition of translation elongation. Subsequently, mitochondrial protein expression is reduced leading to mitochondrial dysfunction, which is sensed by OMA1-DELE-1, and activation of HRI. Additionally, unfolded proteins accumulate in the endoplasmic reticulum, inducing the UPR and activating PERK. HRI and PERK converge to phosphorylate eIF2α which inhibits translation initiation and drives cancer cell cycle arrest. Created in BioRender. Harvey, R. (2025) https://BioRender.com/6lko9fz.

used without specific consent according to the provisions of the Human Tissue Act 2004; in summary, use was part of an ethically approved project, the research team were blinded to patient identity, and no *post mortem* tissues were involved.

## Cell culture

A549 (CCL-185), HeLa (CCL-2), MCF7 (HTB-22), MCF10A (CRL-10317) and HT-29 (HTB-38) were obtained from ATCC. WT and S51A MEFs were a gift from David Ron. A549, HeLa, MCF7, HT-29 and MEFs were cultured in Dulbecco's Modified Eagle Medium (DMEM) (Gibco, 41966-029), supplemented with 10% foetal bovine serum (FBS) (Sigma,

F9665). Mesothelioma-derived primary cells (7T, 8T, 9T and 13T) were derived in previous studies[36,56] and cultured as outlined in Grosso et al.[36], with growth medium consisting of Roswell Park Memorial Institute Medium (RPMI)−1640 (Gibco, 21875-034), supplemented with L-glutamine (2 mM) (Fisher, 25030-081), 100 U/ml penicillin and 100 μg/ml streptomycin (ThermoFisher, 15140122), 20 ng/ml EGF (Peprotech, AF-100-15), 1 μg/ml hydrocortisone (Sigma-Aldrich, H0888), and 10% FBS (Sigma, F9665). MCF10A cells were cultured in DMEM/F12 (Life Technologies, 11330) supplemented with 5% horse serum (Labtech, DH-291/500), 20 ng/ml EGF (Peprotech, AF-100-15), 10 μg/ml insulin (Sigma-Aldrich, I9278), 500 ng/ml hydrocortisone

(Sigma-Aldrich, H0888) and 10 ng/ml cholera toxin (Sigma, C8052). Galactose containing media consisted of glucose-free DMEM (Life Technologies, Gibco) supplemented with 10 mM galactose, 2 mM L-glutamine, 1 mM sodium pyruvate and 10% FBS. All cell lines were incubated at 37 °C and 5% $CO_2$. Details on the compounds used in this study are present in Supplementary Table 1.

### Cell growth analysis—crystal violet and xCELLigence proliferation assay

Cell proliferation was assessed through crystal violet staining or real-time monitoring using xCELLigence. For Crystal violet staining, cells were seeded in 6-well plates at a density of 50,000 cells per well. Subsequently, cells were treated with GC7 as described, fixed in 4% paraformaldehyde (PFA) (ThermoFisher, 043368.9 M) for 10 min at 25 °C, stained with 0.05% crystal violet, rinsed with water, and air-dried at 25 °C. The absorbance at 540 nm was quantified using a Power Wave XS2 plate reader (BioTek). For real-time monitoring of cell proliferation, we used the Agilent xCELLigence real-time cell analysis (RTCA) DP platform. This instrument utilises culture plates with embedded microelectrode sensors that allow label-free quantification of cell proliferation, morphology, and adhesion to the plate. Initially, 5,000 cells are seeded in each well, and the next day, the cell media is replaced, followed by treatment with the respective drug or siRNA transfection mix. Throughout a period of up to 7 days, the machine continuously records changes in impedance measurements, which are provided as an arbitrary value termed cell index, that can be inferred as cell proliferation. For visualization and data analysis, we utilized the RTCA software developed by ACEA Biosciences.

### Immunoblotting

After the experimental treatments, cells were subjected to PBS washes and then lysed in Radioimmunoprecipitation assay (RIPA) buffer (50 mM Tris-HCl pH 7.4, 150 mM NaCl, 1% Triton X-100, 0.1% SDS and 0.5% sodium deoxycholate) supplemented with cOmplete™ EDTA-free protease inhibitor cocktail (Merck, 11697498001), PhosSTOP™ (Merck, 4906845001), and Benzonase® Nuclease (Millipore, 70746) at a specified concentration of 0.2 U/ml. Protein concentration quantification was accomplished using the Pierce™ BCA Protein Assay Kit (Thermo Fisher Scientific) following the manufacturer's guidelines. Subsequently, protein lysates were resolved on NuPAGE™ 4–12% gels (ThermoFisher, NP0321BOX). Electrophoresis was carried out in 1X MOPS running buffer at 150 V for approximately 1.5 h. Proteins were transferred to an Immobilon®-FL PVDF Membrane (Millipore, IPFL00010) and membranes were blocked using 5% milk in Tris-buffered saline, 0.1% Tween-20 (TBST). Membranes were incubated with primary antibodies overnight at 4 °C and incubations with secondary antibodies were carried out for 1-h at room temperature. Membranes were imaged using the Odyssey infrared system (LI-COR Biosciences). Comprehensive details about the antibodies utilized can be seen in Supplementary Table S2. Densitometry analysis was carried out using Image Studio Lite. All signal for phosphorylated proteins were normalised to the total abundance of the corresponding protein. All non-phosphorylated proteins were normalised relative to Actin.

### Immunofluorescence and confocal microscopy

A549 cells were cultured on a 1.5 mm glass coverslip within a six-well culture plate at a density of 50,000 cells per well. Following the specified durations of GC7 treatments, cells underwent PBS washing and fixation in 4% formaldehyde (Sigma-Aldrich, F8775) for 10 min at room temperature (RT). Subsequently, permeabilization was conducted using PBS-0.2% Triton X-100 for 30 min at RT. Following a blocking step in 3% bovine serum albumin (BSA) (Sigma-Aldrich, A-7906) in PBS supplemented with 0.1% Triton X-100 (PBST), cells were subjected to

overnight incubation with anti-mitochondrial protein TIMM44 antibody at 4 °C. After washing in PBST, cells were further incubated with Goat anti-Rabbit IgG (H + L) Highly Cross-Absorbed secondary antibody Alexa Fluor™ 488 (Invitrogen, A-11034) for 1 h at 25 °C. Nuclei were stained with Hoechst dye for 5 min at 25 °C (Invitrogen, 33258) followed by additional washes with PBST. Finally, cover slips were mounted on slides using ProLong Diamond Antifade Mounting Media (Invitrogen, P36970). Confocal images were acquired using an inverted Zeiss LSM 880 confocal with Airyscan using the Plan-Apochromat 63x/1.4 Oil DIC. Image acquisition was performed using Zen Black software (Zeiss). Mitochondrial morphology (elongated or fragmented) was assessed in 80 cells per condition per biological repeat. Image analysis was carried out using Fiji software.

### Flow cytometry

All experiments were conducted using BD LRS Fortessa or BD FACS Canto II flow cytometers. A total of 10,000 events were acquired for each condition and data was analyzed using FlowJo data analysis software (Version 10.10). For quantification of cell death, cells were resuspended in Annexin binding buffer (BD Bioscience) and incubated with Annexin V-FITC (Invitrogen, BMS500FI-300) and Draq7 (Abcam, ab109202) for 20 min at room temperature. For cell cycle analysis, cells were trypsinised, pelleted and fixed in ice cold 70% ethanol in PBS overnight at 4 °C. Fixed cells were incubated with either FxCycle Violet DNA stain (4,6-diamidino- 2-phenylindole dihydrochloride) (Thermo Fisher Scientific) in 0.1% BSA/PBS overnight at 4 °C, or 20 µg/ml propidium iodide (PI) in 1% Triton X-100/PBS and 0.1 mg/ml RNase A overnight at room temperature. Cell cycle state was determined using Dean-Jett-Fox modelling[57].

### TMA staining and quantification

Tissue microarrays (1 mm cores) were constructed from formalin fixed paraffin embedded (FFPE) human resected primary Lung adenocarcinoma from the LATTICeA cohort. 4 µm sections were mounted on to TOMO hydrophilic adhesive microscope slides (Matsunami).

Automated multiplex immunofluorescent staining was performed on the Ventana Discovery Ultra platform (Roche Tissue Diagnostics, RUO Discovery Universal V21.00.0019) using a fluorophore tyramide-based signal amplification system (Akoya Biosciences). Slides were baked on board at 60 °C, followed by a dewax step for 24 min at 69 °C, and heat induced epitope retrieval was applied for 32 min at 95 °C using a Ph9 solution (CC1, Roche Tissue Diagnostics, 950–500). Discovery Inhibitor (Roche Tissue Diagnostics, 760-4840) was applied for 12 min to block any previous endogenous peroxidase. The primary-secondary- antibody pairs used in the multiplex panel alongside their assigned opal fluorophore were applied in a sequential manner, respectively, are as follows: pan-CKAE1-3 (Leica Biosciences, NCL-L-AE1-3, 1:250, 28 min), OmniMap-anti Ms HRP (Roche Tissue Diagnostics, 760–4310, 12 min), Opal 620 (Akoya Biosciences, OP001004, 1:100, 8 min); Hypusine (Sigma-Aldrich, ABS1064-I, 1:100, 1 h), Opal 520 (Akoya Biosciences, OP001001, 1:400, 8 min); Ki67 (30-9) (Roche Tissue Diagnostics, 790-4286, RTU, 20 min), OmniMap-antiRb HRP (Roche Tissue Diagnostics, 760–4311, 12 min), Opal 570 (Akoya Biosciences, OP001003, 1:200, 8 min); eIF5A (Abcam, ab137561,1:500, 32 min), OmniMap-antiRb HRP (Roche Tissue Diagnostics, 760–4311, 12 min), Opal 690 (Akoya Biosciences, OP001006, 1:300, 8 min); SDHB (EPR10880), (Abcam, ab175225, 1:2000, 32 min), OmniMap-antiRb HRP (Roche Tissue Diagnostics, 760–4311, 12 min), Opal 650 (Akoya, OP001005, 1:300, 8 min). Following each detection with opal throughout the multiplex sequence, a de-crosslinking step was included to denature any previous antibody binding using a heat-and -pH mediated approach at 100 °C using a pH 6 solution for 24 min (CC2, Roche Tissue Diagnostics, 950–123). Whole slide images were scanned on the PhenoImager multispectral slide scanner (Akoya Biosciences, version V1.0.13) using a 10× objective. Regions of interest were

selected using Phenochart (Akoya, Biosciences) (v1.1.0) and individual core regions of interest were acquired at 20× for further analysis. Core images were spectrally unmixed using Inform (Akoya Biosciences) (version 2.6.0). Visiopharm was used for all image analysis. For tissue segmentation, a bespoke deep learning algorithm (version 2023.01.3.14018) was trained on annotated images verified by a pathologist using the deep learning module U-Net and DAPI, Cyto-keratin, and autofluorescence features to segment tumour, stroma, and necrosis regions of interest (ROIs). For cell detection, an additional deep learning algorithm (version 2022.12.0.12865) was trained using Dapi and Ki67 inputs to generate background, boundary, and nuclear features to detect nuclei and to generate Ki67+ and Ki67- nuclear classes for output. Post processing steps were included to remove background and separator labels, and to turn the Ki67+ and Ki67- labels to a single nuclear label. Cytoplasmic labels were generated by dilating nuclear labels by 20 and 10 pixels for tumour and stromal cells, respectively. Feature heatmaps were created to separate nuclear and cytoplasm objects where cells were merged. Output variables were generated for ROI area, mean pixel intensities for each marker, and X Y coordinates for each cell and were exported for each core image for subsequent statistical analysis. Measures obtained from Visiopharm were analysed using R (version 4.2.2) with tidyverse packages (version 2.0.0). Stained tissue sections have been preserved.

### Puromycin incorporation assay

Cells were treated with 10 μg/ml puromycin for 10 min before being harvested and lysed in RIPA buffer. Subsequently, the samples underwent immunoblotting, as detailed previously. Loading controls were carried out using Coomassie blue gel staining or Ponceau S membrane staining. For Coomassie blue, gels were stained with InstantBlue® Coomassie Protein Stain (Abcam, ab119211) for 15 min. For Ponceau S, membranes were incubated with Ponceau S for 5 min at room temperature and washed 3 times before imaging. Analysis was performed using Image Studio Software v.5.2.

### Polysome profiling

For polysome profiling, an initial seeding of $1 \times 10^6$ cells was carried out in a 15 cm plate. Subsequently, cells were allowed to adhere and grow for the appropriate duration. Following this, the media was changed, and the cells were treated with GC7 for the designated period. Prior to harvesting, the cells underwent a wash with ice-cold PBS supplemented with cycloheximide at 100 μg/ml. The cells were then lysed in buffer containing 100 mM NaCl, 5 mM MgCl$_2$, 15 mM Tris-HCl pH 7.4, 0.2 M sucrose, 0.5% IGEPAL, 100 U RNasin, 1 mM DTT, and 0.1 mg/ml cycloheximide. Protein content was quantified using the BCA protein quantification kit (ThermoFisher, J63283). Approximately 2–2.5 mg of protein per sample was loaded onto sucrose gradients (10–50%) and subjected to centrifugation at $247,767 \times g$ for 2 h at 4 °C. The gradient was displaced from the bottom of the tube using a density gradient fractionation system with continuous monitoring of the absorbance at 254 nm (Teledyne ISCO).

### Ribosome run-off assay

Cells were seeded and treated as for standard polysome profiling assays. However, after treatment with GC7 for 3 h, both control and GC7-treated cells were additionally exposed to 2 μg/ml harringtonine (Supplementary Table 1), to inhibit ribosome loading[58], for either 0, 3, 4, or 5 min. Cells were treated with 100 μg/ml cycloheximide for 3 min and lysed in polysome profiling lysis buffer. Samples were subjected to polysome profiling analysis and polysome density was quantified as previously described[59] by calculating the area under the curve. Ribosome run-off was determined by comparing polysome density from untreated and GC7 treated cells following treatment with harringtonine for 5 min.

### Ribosome Stalling Assay

Stalling assay was carried out as described in ref. 60. After exposure to 0.2 μM anisomycin for 15 min and 10 μM GC7 for 3 h, respectively, cells were collected in lysis buffer containing 20 mM HEPES pH 7.5, 100 mM NaCl, 5 mM MgCl$_2$, 100 μg/ml digitonin, 100 μg/ml cycloheximide, cOmplete mini EDTA-free protease inhibitor cocktail (Merck, 11697498001), and 200 U RNase inhibitor (NEB, M0307L). The lysates were incubated at 4 °C for 5 min and subsequently centrifuged at $17,000 \times g$ for 5 min at 4 °C. Then, 1 mM calcium chloride was introduced, and the samples underwent digestion with 500 U micro-coccal nuclease (NEB, M0247S) at 22 °C for 30 min. Following this, 2 mM EGTA was added to terminate the reaction. Approximately 2–2.5 mg of protein per sample were applied to a 10–50% sucrose gradient, which underwent centrifugation at $247,767 \times g$ for 2 h at 4 °C. Finally, the gradients were processed through an ISCO density gradient fractionation system, with the absorbance continuously measured at 254 nm.

### qPCR

To validate the efficacy of OMA1 and DELE1 knockdown following siRNA transfections, quantitative PCR (qPCR) analysis was employed. Specific primers for OMA1 and DELE1 mRNA were designed (Supplementary Table 3). After siRNA transfections, cells were harvested in Trizol Reagent (ThermoFisher, 15596026), and RNA extraction was performed following the manufacturer's protocol. RNA concentration was determined using a Nanodrop 2000 (Thermo Scientific), and 1 μg of RNA per sample was utilized for cDNA synthesis with Superscript™ III Reverse Transcriptase (ThermoFisher, 18080093), following the manufacturer's instructions. Subsequent qPCR was conducted using SensiFAST SYBR Lo-ROX Kit (Meridian Bioscience, BIO-94020) on the QuantStudio™ 6 Flex (Thermo Scientific).

### siRNA transfections

For siRNA transfection, ON-TARGETplus™ siRNAs from Horizon (Supplementary Table 4) were used, along with Lipofectamine™ RNAiMAX transfection reagent (ThermoFisher Scientific, 13778075), following the manufacturer's protocol. Cells were initially seeded at a density of 50,000 cells per well in a 6-well plate or at 5000 cells/well in a 16-well xCELLigence plate. The following day, cells were supplemented with the appropriate transfection mix and further incubated for 18 h. Subsequently, the media was changed, and the cells were allowed to grow for an additional 24–72 h, depending on the specific experiment. Following this, the cells were treated with GC7 and either subjected to growth monitoring in an xCELLigence proliferation assay or harvested for western blot or qPCR analysis, as described previously.

### ROS detection assay

For ROS detection we used ROS-Glo™ H$_2$O$_2$ Assay (Promega, G8820), according to manufacturer's protocol. Menadione (Supplementary Table 1) was used as a positive control.

### ATP assays

CellTiter-Glo® 2.0 Cell Viability Assay (Promega, G9241) was used to quantify ATP levels according to manufacturer's protocol. Rotenone was used as a positive control.

### Extracellular flux analyses

Real-time oxygen consumption rates (OCR) were measured using a Seahorse XF96 extracellular flux analyser (Agilent). A549 ($2 \times 104$) and MCF10A ($3 \times 104$) cells were seeded in glucose containing media in Seahorse Biosciences XF96 plates. Cells were washed in unbuffered serum-free DMEM Seahorse Assay Medium and incubated in a non-CO$_2$ 37 °C incubator for 1 h before placing in the Seahorse XFe96 Analyzer. Oligomycin (2 μM), FCCP (500 nM), rotenone/antimycin A (2 μM) were

added sequentially. Mean OCR values were obtained from a minimum of four technical replicates and normalised to total protein content.

## Dynamic stable isotope labelling by amino acids in cell culture (SILAC) mass spectrometry

Dynamic SILAC was conducted using the SILAC Protein Quantitation Kit (Trypsin) that contained media supplemented with either light or heavy labelled lysine and arginine. A total of $1 \times 10^6$ cells were seeded in a 15 cm plate and allowed to attach and grow for 48 h under standard growth conditions, as detailed earlier. Subsequently, the media was replaced with either normal media in the control cells or media enriched with heavy labelled $^{13}C_6$ $^{15}N_2$ L-lysine-2HCl and $^{13}C_6$ $^{15}N_4$ L-arginine-HCl for the same duration as the GC7 treatments. Subsequently cells were harvested in 25 mM AMBIC containing 0.2% Rapi-Gest detergent (Waters). 50 µg of total protein of each sample was diluted to 130 µl total volume using 25 mM AMBIC containing 0.1% RapiGest (w/v) (Waters). The cysteines were reduced by adding DTT to a final concentration of 4 mM and then alkylated by adding iodoacetamide to a final concentration of 14 mM. Proteins were digested by adding 1 µg trypsin (Promega, UK) for 18 h at 37 °C. Peptides were acidified with the addition of trifluoroacetic acid to a final concentration of 0.5% (v/v). The acidified tryptic digest was then centrifuged at $13,000 \times g$ for 15 min and the supernatant subjected to LC-MS/MS analysis.

LC-MS/MS was performed on an Ultimate U3000 HPLC (Thermo-Fisher Scientific, San Jose, USA) hyphenated to an Orbitrap Eclipse mass spectrometer (ThermoFisher Scientific, San Jose, USA). Peptides were trapped on a C18 Acclaim PepMap 100 (5 µm, 300 µm × 5 mm) trap column (ThermoFisher Scientific, San Jose, USA) and eluted onto a 50 cm EasySpray column (ThermoScientific Dionex, San Jose, USA) using 150-min gradient of acetonitrile (5–35%). For data dependent acquisition, a dual FAIMS compensation voltage experiment was performed (−45v, −65v), with a cycling time of 1.5 s between the two experiments. MS1 scans were acquired at a resolution of 120,000 (AGC target of 4e5 ions with a maximum injection time of 50 ms), followed by MS2 scans acquired at a resolution of 15,000 (AGC target of 5e4 ions with a maximum injection time of 25 ms) using a collision-induced dissociation energy of 30 and an isolation window of 0.7. Dynamic exclusion of fragmented m/z values was set to 60 s. Raw data were imported and processed in Proteome Discoverer v2.5 (Thermo Fisher Scientific). The raw files were submitted to a database search using Proteome Discoverer with Sequest HT against the UniProt reference proteome for Homo sapiens using SILAC quantification method for both heavy lysines and arginines. The processing step consisted of a double iterative search using INFERIS Rescoring algorithm on a first pass with methionine oxidation set as variable and cysteine carbamidomethylation, heavy arginine and lysine as fixed modifications. For the second pass, all spectra with a confidence filter worse than high were researched with Sequest HT including additional common protein variable modifications [Deamidated (N, Q); gln to pyro-Glu (Q); N-terminal acetylation and Methionine loss). The spectra identification was performed with the following parameters: MS accuracy, 10 ppm.; MS/MS accuracy of 0.02 Da; up to two trypsin missed cleavage sites were allowed. Percolator node was used for false discovery rate estimation and only rank 1 peptide identifications of high confidence (FDR < 1%) were accepted. Subsequent data analysis procedures were executed as described below.

## Proteomics data analysis

Source code for the proteomics data analysis to perform all the analyses below is available from https://github.com/MRCToxBioinformatics/eIF5A_hypusination_inhibition_silac v0.1. Peptide-level quantifications from Proteome Discoverer were processed to obtain protein-level newly synthesized protein proportions. Peptides matching contaminant proteins or multiple master proteins were excluded. The heavy and light peptide intensities were used to calculate the proportion of nascent peptide as heavy/(light + heavy). Peptide-level proportions were then aggregated to protein level by taking the median proportion. Proteins with only one peptide were excluded.

Differential abundance of newly synthesized proteins was performed using DEqMS. Since DEqMS expects a Gaussian-distributed abundance, we converted the proportions back to a ratio of heavy and light intensities, where the ratio is calculated as proportion/(1-proportion). Ratios were log2-transformed to make them Gaussian distributed. The DEqMS model was ratio - condition + replicate, where the replicate was a blocking factor, and the condition was GC7 or control. DEqMS models the relationship between 'peptide count' and variance to share information between proteins. We followed the developer's advice and used the minimum number of peptides per protein across the replicates with a quantified ratio as the peptide count. Only proteins with at least two quantified ratios in both conditions for a given time point could be tested. To identify proteins with a significant change in the newly synthesized protein ratio, we used the adjusted p-value calculated by DEqMS and a 5% False Discovery Rate (FDR) threshold[61].

The identification of over-represented Gene Ontology (GO) terms was performed using goseq, which enables adjustment for bias towards proteins for which there was more statistical power to detect a significant difference. The peptide count used with DEqMS, as described above, was provided as the bias factor. P-values were adjusted for multiple testing using the False Discovery Rate procedure[61]. GO terms over-represented with an adjusted p-value less than 0.01 (1% FDR) were taken as significant. GO terms with fewer than 10 proteins in the foreground were excluded. Following this, redundant GO terms were removed, and the magnitude of the over-representation was estimated using the functions remove_redundant_go and estimate_go_overrep from camprotR v0.0.0.9000.

The complete human Mitocarta3.0 database table was downloaded on 26 July 2023. This denoted the sub-mitochondrial localisation of each protein. Proteins with multiple localisations were excluded from the visualisations. We used the uniprot_map function from uniprotREST to obtain the amino acid sequence for each protein and calculated the frequency of each amino acid and di/tri-peptide. We used binomial ridge regression to determine the amino acid composition features which were predictive of a significant loss of newly synthesized protein at 6 h. This was performed using the cv.glmnet function from the glmnet package.

## Reporting summary

Further information on research design is available in the Nature Portfolio Reporting Summary linked to this article.

## Data availability

The mass spectrometry proteomics data generated in this study have been deposited in the ProteomeXchange Consortium via the PRIDE[62] partner repository and are accessible with the dataset identifier PXD051303. LATTICeA raw images can be requested from JLQ upon reasonable request. Clinical data from the LATTICeA cohort is restricted, and is held by NHS Greater Glasgow and Clyde Biorepository (clare.orange@ggc.scot.nhs.uk; john.lequesne@glasgow.ac.uk) as custodians. Data access can be requested, and any such request will be reviewed and released under their research ethics committee-approved tissue bank protocols. Requests will be reviewed and approved within 6–8 weeks and will be accompanied by a data sharing agreement detailing the conditions and restrictions of use and publication. Source data are provided with this paper.

## Code availability

Source code for proteomics data analysis is freely available from https://github.com/MRCToxBioinformatics/eIF5A_hypusination_inhibition_silac v0.1.

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

## Acknowledgements

We thank the Flow Cytometry, Microscopy, Mass Spectrometry and Bioinformatics facility of MRC Toxicology Unit for their support throughout this study. A.P.S., is supported by is supported by Cancer Research UK Technology, UK Grant G108392, A.E.W. M.S., R.F.H., T.S. and A.R., are supported by the Medical Research Council, grant number MC_UP_A600_1023 (A.E.W.) T.E. M acknowledges funding from Wellcome Leap as part of the R3 Program. MB was funded by CRUK A29252 and A31287, OJS was funded by CRUK DRCQQR-May21\100002, A24388 and A31287, JPCLQ is funded by the Mazumdar-Shaw Molecular Pathology Chair endowment at the University of Glasgow, and CRUK EDDPGM-Nov21\100001. We also thank NHS Greater Glasgow and Clyde Biorepository, the Glasgow Tissue Research Facility, and the CRUK Scotland Institute Histology Facility. We would like to thank Rachel Grimley, Neil Jones, Nathan Breeds, Sheena Patel and Azadeh Bagherzadeh from Cancer Research Horizons for helpful discussions.

## Author contributions

Conceptualization: A.E.W., A.P.S., M.B., M.S., R.M.R., R.S., O.J.S. and M.T.; methodology: A.P.S., M.S., R.F.H., T.S. and J.P.C.L.Q.; investigation: A.P.S., R.M.R., T.S., T.E.M., M.P., V.D., A.R., C.H.C., R.L.P., I.P., L.O.J., R.F.H., X.M.S., K.A.D., H.A.N. and M.M.; writing (original draft): A.P.S., A.E.W., R.F.H., M.S. and T.S.; writing (review and editing): A.E.W., A.P.S., T.S., M.S., R.F.H., M.B., J.P.C.L.Q., O.J.S., R.L.G. and R.S.; resources: A.P.S.; data curation: T.S. and A.P.S.; supervision: A.E.W., A.P.S., M.B., O.J.S., J.P.C.L.Q. and M.T.; project administration: A.E.W.; funding acquisition: A.E.W., O.J.S., M.B., J.P.C.L.Q. and M.T.

## Competing interests

The authors declare no competing interests.

## Additional information

Aristeidis P. Sfakianos[1,5] ✉, Rebecca M. Raven[1,5], Tom Smith[1], Xiao-Ming Sun[1], Thomas E. Mulroney[1], Mariavittoria Pizzinga[1], Veronica Dezi[1], Angela Rubio Tenor ®[1], Mark Stoneley[1], Cameron H. Cole[1], Karam Al-Doori[1], Hashim Ahmed Nur[1], Rachel L. Pennie[2], Ian Powley[2], Leah Officer-Jones ®[2], Ritwick Sawarkar ®[1], Martin Turner ®[3], Marion MacFarlane[1],

**Owen J. Sansom** ⓘ [2,4], **Martin Bushell**[2,4], **John Le Quesne** ⓘ [2,4], **Robert F. Harvey** ⓘ [1] ✉ **& Anne E. Willis** ⓘ [1] ✉

[1]MRC Toxicology Unit, University of Cambridge, Cambridge, UK. [2]Cancer Research UK Scotland Institute, Glasgow, UK. [3]The Babraham Institute, Babraham Research Campus, Cambridge, UK. [4]School of Cancer Sciences, University of Glasgow, Glasgow, UK. [5]These authors contributed equally: Aristeidis P. Sfakianos, Rebecca M. Raven. ✉e-mail: arissfak92@gmail.com; rfh32@mrc-tox.cam.ac.uk; aew80@mrc-tox.cam.ac.uk

