## [Transparent Peer Review file · Nature Communications]

eIF5A-dependent feedback inhibition from mRNA translation elongation to initiation limits tumour cell proliferation

Corresponding Author: Professor Anne Willis

Version 0:

Reviewer comments:

Reviewer #1

(Remarks to the Author)

In this work, Sfakianos and colleagues studied the effects and molecular consequences of EIF5A inhibition in tumor cells. Through a comprehensive approach based on several technologies, they propose that inhibition of EIF5A hypusination causes a bi-phasic translational response characterized by an early inhibition of protein elongation, which feeds back on a later eIF2alpha-mediated inhibition of translational initiation, and consequent decrease of global protein synthesis. This mechanism would impair mitochondrial function and induce apoptosis in tumor cells characterized by high OXPHOS, such as A549 cells.

The proposed model of an elongation-to-initiation feedback mechanism provides novel information to the EIF5A/translation field. However, there are major limitations in this study that dampen the enthusiasm for the whole story:

1) Approach: the work is almost entirely based on the use of the DHPS inhibitor GC7 to inhibit EIF5A function, mostly using one cell line (A549).

It is generally believed in the field, and documented in some reports (e.g. Oliverio et al. *Amino Acids* 2014 DOI 10.1007/s00726-014-1821-0; Landau et al., *JBC* 2010 <https://doi.org/10.1074/jbc.M110.106419>), that GC7 has effects unrelated to DHPS inhibition and may directly target mitochondrial proteins. Even if lower doses and specific cell lines have been used in this work, authors have not provided evidence of the specificity of GC7 under their conditions. In Figure S3 it is shown that the knockdown of both EIF5A1 and 2 induces markers of ISR and decrease of translation rate. However, the authors have knocked down EIF5As, but not DHPS, the actual target of GC7. To demonstrate the time-dependent effect of decreased/abrogated EIF5A hypusination, the authors should use inducible DHPS knockdown and/or cells stably expressing a mutant EIF5A lacking the hypusinated lysine (i.e. EIF5AK50R) in all the experiments shown in the manuscript. As the paper stands now, only the consequences of GC7 treatment are carefully documented, but this is not a trustable reflection of decreased EIF5A hypusination and function.

2) Previous literature/novelty: Previous publications not mentioned by the authors have addressed issues overlapping with this paper, sometimes providing evidence in conflict with the present model.

a) The group of Chaim Kahana described the effect of EIF5A inhibition on translational initiation (Landau et al., *JBC* 2010 <https://doi.org/10.1074/jbc.M110.106419>). Using mouse fibroblasts, they observed that treatment with GC7 inhibited translational initiation and induced eIF2alpha phosphorylation, although with kinetics that were different from this submitted paper. Indeed, they saw an early (from 3h) increase of phosphorylated eIF2alpha (Fig 6B), in contrast with the present manuscript, where the authors claim a later (24h) activation of this protein. Of note, upon interference of EIF5A, Kahana and collaborators failed to reproduce the GC7-mediated decrease of initiation (Fig 4), whereas they noticed decreased elongation. This discrepancy was attributed to non-specific effect of GC7.

Authors (who have also used mouse fibroblasts, Fig 1E) need to address these conflicting data.

In another paper (Sun et al., *J Cell Physiol.* 2010 Jun;223(3):798-809.doi: 10.1002/jcp.22100) it was shown that deletion of EIF5A prevents apoptosis, while its overexpression induces cell death, causing activation of the intrinsic mitochondrial pathway. Therefore, the observations are in stark contrast with the submitted report, where EIF5A inhibition with GC7 induces apoptosis. Of importance, Sun et al. showed that the cytotoxic effect of EIF5A was not due to hypusination since overexpression of hypusination-defective EIF5A mutants still induced apoptosis (Fig 2 and 3). This conflicting data should also be carefully addressed and taken into account when planning the experiments.

3) Clinical relevance: Another major limitation of this study is the lack of in vivo evidence. A central claim of this paper is that hypusinated EIF5A inhibition may represent a plausible novel strategy to target tumors dependent on mitochondrial function.

However, only in vitro studies are shown (Fig 6). It would be important to provide solid evidence of the pathophysiological importance of these findings using relevant in vivo models of tumors harboring inducible DHPS and/or EIF5AK50R mutants to recapitulate the in vitro observations.

Additional Points

- 1- The representative Figure 1A looks different from the quantifications shown on the side (B-F).
- 2- The decrease of hypusination (1A-B) after 0.5-3-6 hrs is quite modest (20-25%). It is difficult to concur that it might be biologically relevant.
- 3- The downregulation of total EIF5A after longer GC7 exposure (1A-C) is also concerning and not well explained.
- 4- The claimed temporal changes of p-eEF2 and p-eIF2alpha and ATF4 are not very convincing: judging from figures 1A, E, F, variable and subtle differences of questionable relevance seem to occur in the earlier time points.
- 5- Data in Fig S1 are also not convincing. The early increase of puromycin incorporation in Fig 1G does not look very strong.
- 6- Figure S3: only one band is shown for EIF5A1 and 2. These proteins migrate differently and typically form a doublet. Do A549 cells express both isoforms? In this case, both bands should be detected.
- 7- Figure S4A why 40 uM of GC7 was used in HeLa cells and 10 uM in A549? Did the authors perform dose-response studies?
- 8- Figure 6A-B the effect of GC7 on MCF10A is defined as non-significant but the proliferation rate shown in Figure 6B seems reduced by GC7. Perhaps the number of replicates should be increased.

Reviewer #2

(Remarks to the Author)

In this article, Sfakianos and colleagues provide evidence suggesting that the pharmacological inhibition of eIF5A hypusination triggers eIF2alpha phosphorylation and suppresses global protein synthesis. The authors then took advantage of dynamic SILAC to identify the alterations in the proteome enticed by suppression of eIF5A activity, and on a selected subset of targets distinguished those that were mediated by feedback induction of ISR. The authors also provided some evidence that the mechanisms of induction of eIF2alpha phosphorylation upon eIF5A inhibition may be the consequence of activation of the UPR/PERK axis and impaired mitochondrial function which appears to be mediated by OMA1 and DELE1 proteases thus leading to activation of HRI. Finally, the authors show that the pharmacological inhibition of eIF5A hypusination leads to cancer cell death which appears to at least in part be mediated by ISR induction. Overall, it was thought that this report is of broad interest to mRNA translation and cancer research communities since it puts forward a previously unappreciated model of reprogramming of protein synthesis and cell death induction triggered by the inhibitor of eIF5A hypusination. It was thought that the most of applied methodology was appropriate. In general, provided data were thought to adequately support author's conclusions. Nonetheless, several concerns were raised as outlined below whereby it was thought that addressing them may further solidify this study.

Major comments:

-Throughout the manuscript, it remained rather unclear how densitometry was performed (e.g., what was used to normalize the data). Considering that the article heavily relies on this analysis, this information should be provided in each figure legend.

-Line 455: "Western blot analysis of puromycin incorporation revealed an augmented incorporation in the early stages of GC7 treatments, which is likely indicative of an accumulation of slow-moving ribosomes on the mRNAs and a reduction in elongation". I am a bit confused by this statement, as it is somewhat unclear to me how slowed elongation may result in the induction of puromycin incorporation. Can the authors elaborate on this? These effects appear to parallel the induction of ISR and thus the alternative explanation may be that the shut-down of global protein synthesis may at least in part occur via induction of ISR, while the effects of modest inhibition of eIF5A hypusination (early time points) on global mRNA translation may be marginal. Although authors speculate that relatively late downregulation of global protein synthesis upon GC7 treatment may be due to inhibition of initiation, it appears warranted to test the effects of GC7 treatment on global protein synthesis in S51A eIF2alpha KI MEFs. Finally, the technical quality of the figure 1G is not the greatest and should likely be improved.

-Polysome profiles presented in figure 1I may not be of sufficient resolution to substantiate the authors claim that 80S peak remains unchanged upon GC7 treatment. Namely, in the provided polysome tracings 80S peak appears to be unidentifiable.

-Quantification in S3H does not seem to match blot in S3G.

-The relationship between tested proteins in Figure 3 and proteome analysis should be specified. Of note, it was hard to find out actual proteins that were affected in SILAC study as well as raw data associated with it. Were OMA1 and DELE1 levels affected by GC7? Also, the 1/2 lives of ribosomal proteins are notoriously long and thus the authors may want to check what they may be for rpS6 and rpS25 as this may mask the effects on de novo synthesis of these proteins.

- A549 and MCF10A cells are highly genetically heterogenous. Thus, it was thought that it would be warranted to include isogenic non-transformed and transformed cell lines in order to determine whether the effects of GC7 are specific to cancer

cells and/or tone down associated claim.

-It appears somewhat ambiguous to define A549 cells as those having "high OXPHOS" levels. What was the criteria for this? When A549 cells were cultured in galactose containing media, the ATP production was decreased (Figures 6F and G) which suggests that their mitochondria are actually dysfunctional, and unlikely to exhibit "high OXPHOS" levels. Moreover, when A549 cells were forced to rely on mitochondrial metabolism by culturing them on galactose without glucose exhibited enhanced sensitivity to GC7 (Figure 6H). Considering this, it was thought that the authors may want to reconsider whether the sensitivity to GC7 is linked to "high OXPHOS levels" or perhaps on mitochondrial dysfunction which appears to be more likely from the aforementioned experiments. Finally, it appears warranted to include some respirometry studies and determine bioenergetic profiles of investigated cell lines as well as the effects of GC7 on bioenergetics +/- ISRIB.

Minor issues:

-More comprehensive introduction of ISR may be warranted.

-The article needs some polishing when it comes to consistency and editing. In addition, the figure legends were found to be somewhat incomplete whereby some important details (e.g., representation of the error bars i.e., SD or SE), number of replicates in some panels are missing, etc.) appear to be missing.

-Alternative genetic approaches (e.g., depletion of DHPS) to parallel GC7 studies may be more appropriate than depleting eIF5A1/2.

-Although the immunogen used to generate eIF5A Ab appears to be proprietary, the authors should specify whether this antibody recognizes both eIF5A1 and 2. Alternatively, RT-qPCR should be used to validate the level of KD of each isoform by RNAi.

-Some explanation should be provided why the cells were pretreated with ISRIB, and why ISRIB was not applied after GC7 if the inhibition of eIF5A hypusination precedes ISR.

-The role of mTOR in the observed phenomena remains unclear. This appears to be pertinent as GC7 appears to affect mTOR targets implicated in elongation (e.g., eEF2K, albeit this may be AMPK too) and initiation (e.g., 4E-BPs). Albeit it is understandable that this may be out of the scope of this study, this should be reflected in discussion section.

Out of curiosity:

Are the levels of DOHH sensitive to mTOR inhibition and/or are they known to be regulated by ATF4?

I hope that the authors will find these comments helpful and of sufficient pathos.

Sincerely

I/Topisirovic

Reviewer #3

(Remarks to the Author)

eIF5A-dependent feedback inhibition from mRNA translation elongation to initiation induces apoptosis in tumor cells by Sfakianos and colleagues

this manuscript reports investigations into the action of a well-characterized eIF5A-directed inhibitor called GC7 on protein synthesis, mitochondrial function and cell viability in cancer-derived cell lines, particularly A549 cells, a lung adenocarcinoma cell model. eIF5A and its poorly expressed isoform eIF5A2 are the only proteins post-translationally modified with hypusine, a polyamine derived from spermidine. GC7 inhibits DHPS, the enzyme that performs the first step of hypusination, leading to poorly functioning eIF5A. Here GC7 addition impacted both hypusination and levels of eIF5A.

eIF5A binds vacant ribosome E sites and stimulates peptide bond formation between A and P-site bound aa-tRNAs, thus promoting elongation, including formation of the first peptide bond. eIF5A can also enhance translation termination and has been shown by structural studies to be associated with stalled ribosomes/disomes and RQC intermediate complexes. Although impact on elongation is made clear, in this study there are no assays to specifically examine any potential impact of GC7 on termination.

The study provides evidence that GC7 treatment leads to activation of other translational control pathways that act to diminish global protein synthesis. This makes mechanistic sense as other studies are consistent with feedback control systems operating to curtail protein synthesis in various ways when one later step is limiting. In recent years GCN2 has been a focus for studies showing ribosome stalling activates eIF2 phosphorylation. In contrast in A549 cells it appears GCN2 is not involved following GC7 treatment and instead attention is focused on mitochondria and the recently described OMA1/DELE1 mediated activation of HRI. By use of an ISR inhibitor that acts downstream of kinase activation (ISRIB) it is

shown that preventing the feedback loop to dampen initiation can rescue translation of some of the proteins impacted by GC7 (FIG 3). Although it appears from Fig. 3A that only those proteins modestly sensitive to GC7 are rescuable. The way the quantification is shown does not make this easy to verify. Other proteins tested here that appear more severely impacted by GC7 are not rescued by ISRIB at 24 hrs are mitochondrially targeted proteins. When comparing A549 and MCF10A cells the major difference observed was not in translation signalling but in ATP levels, again providing a mitochondrial focus to GC7 sensitivity.

Overall the work identifies and characterizes a feedback loop between the elongation and initiation phases of protein synthesis and shows that this relies in large part on signalling from mitochondria to HRI and eIF2 phosphorylation. The western blots make it clear that other signals are also being relayed from eIF5A, eg via 4E-BP1 and eEF2 that are not further explored here. The work utilizes a wide range of molecular techniques and appears generally carefully performed and interpreted. The main conclusion is that cells relying more heavily on mitochondria including the cancer lines studied here are more susceptible to GC7 than other cells and that it may therefore have some therapeutic value that would require further study to assess. However what the key features of GC7-resistant cells verses sensitive cells is not well explored.

However I have quite a few questions and suggestions regarding the work and its presentation that are outlined below.

Points.

Major

1. The text states that MCF10A were identified as GC7 resistant cells but does not indicate what screen was done.
2. The methods make it clear different media were used to culture A549 and MCF10A cells. Could this change contribute to the observed differential GC7 sensitivity?
3. Western blot quantification. Throughout the MS almost every western it appears that 'lane 1' has been normalized to '1' for all signal quantification. The impact of this is that in the control samples there is no variation around the mean, while there is for all other samples. Thus for the statistics the samples all have unequal variance with the control. Instead of selecting 1 lane to normalize to it is a better practice to normalize all lanes to the mean signal across all the blot lanes. In that way all samples will have meaningful error bars and if there is no variation all lanes will be approximately 1. This permits the stats to be done in a valid way. Implementing this change will impact almost every main figure and many of the supplementary figures.
4. P12 ribosome run off assay. From the data presented it is not obvious how differences in polysome density were calculated. It is clear from Supp Fig 2 that there is a large increase in free 80S upon HH treatment in both control and GC7 cells, but as the 80S peak goes dramatically off scale it becomes difficult to accurately quantify P/M ratios and rule out differential loadings. The methods description does not help the reader here and the presented polysome trace data do not appear different \pm GC7. In panel C should the timescale be in minutes? How the numbers in Fig 1J are derived is also unclear.
5. Line 497. The 'dynamic' SILAC assay could be introduced better as many will not be as familiar with this type of assay. I think using the term 'nascent' to describe heavy labelled proteins that may have been made at anything up to 24hours before the assay point is misleading. In protein synthesis, 'nascent proteins' are typically defined as those that are undergoing synthesis and so are within/exiting the ribosome exit tunnel, not those made hours before. Please consider a different term. What cells were used for this assay? How many replicates are used, the Figure 2A appears to show 2 reps only for GC7 treated cells and 3 for the control, is this correct? If only 2 this will affect the statistical power. Have PCA or other whole dataset comparisons been done to visualise the data and analyze its consistency as a whole?
6. The analysis presented in Supplementary Figure 5D is not at all clear. What is being shown here? What do 0, 1, 2,3 denote on the X axis of each graph? What does AAs: 1-3 mean?
7. What is the source of the ribosome pausing data in Figure s5G?
8. While the SILAC did not find any evidence of poly-proline involvement the authors should acknowledge that they have not sampled the entire proteome. Many poly-proline proteins are poorly expressed, others are secreted, so may not have been captured here. Hence the assay may not have been sensitive enough to capture many poly-proline substrates. Was any western blotting done of well-known poly-proline proteins not covered by the MS analyses?
9. Figure 3 The impact of ISRIB is quite modest, at best ~1.5 fold. It would be good to acknowledge the other impacts seen eg on 4E-BP phosphorylation suggest involvement of mTOR and eg eEF2K indicate wider impacts.
10. Figure 4/Supp Fig 7. siRNA experiments. what cell type used here?
11. The HRI activation model proposed invokes processing and activation of DELE1 by OMA1. Has a change in isoform L-DELE1 to S-DELE1 been observed in MS data with GC7 or via western blotting? This would further support the proposed model.
12. Do GC7 resistant MCF10A cells exhibit the mitochondrial morphology changes seen for A549 cells following GC7 treatment (ie as shown in Figure 5A/B for A549s)

minor

13. Supplementary Fig 1B the graph X axis is out of order 18 hours before 6? Why?
14. Supplementary Fig 1E/H. Was it known previously that p-eEF2 levels are very low in S51A MEFs?
15. Any thoughts as to why eEF2-P signalling is highly responsive to GC7 here?
16. P20 the reason for the switch to HeLa cells is not made clear.
17. Line 533. There are now 6 eIF2 kinases known including MARK2 and FAM69C not studied here.
18. Line 534 eIF2AK2 is wrong here.
19. Supplementary Fig 11. Adding panel numbers and labeling across this figure would help. The stained image text and graph labels are all too small.
20. Line 709 there is no reference 51.
21. Line 713. In the referenced study the polyproline stretch in TIM50 is implicated in eIF5A sensitivity, however while this is

found in yeast TIM50 it is not conserved in human TIMM50, so the mechanisms are likely distinct. This could be clarified in the text.

Reviewer #4

(Remarks to the Author)

The manuscript entitled “eIF5A-dependent feedback inhibition from mRNA translation elongation to initiation induces apoptosis in tumour cells” by Sfakianos et al. is focused on the activity of the translation factor eIF5A and proposes a new pathway of translation regulation.

The authors investigate the use of a well known hypusination inhibitor, GC7, in several cancer cell lines to better understand the molecular mechanism through which this molecule inhibits translation.

Results clearly show that inhibition of eIF5 activity by GC7, but also by knock down with siRNA, has effects not only on translation elongation but also on translation initiation and that the link between eIF5A and translation initiation involves mitochondrial dysfunction, activation of mitochondrial proteases which in turn induce eIF2 phosphorylation by HRI and PERK kinases. The authors claim that this pathway induces apoptosis in some cancer cell types which appear to be eIF5A sensitive and, therefore, suggest that GC7 could represent a potential pharmacological approach to target cancers which are particularly dependent on mitochondrial function for their proliferation.

The manuscript is interesting and presents novel findings shedding light on the mechanism of action of an important translation regulator such as eIF5A. It is well written and well documented in relation to the literature.

In my opinion it deserves publication in this journal with the following revisions:

Line 57: Acronyms (APC) must be specified.

Line 69: The statement “This positioning allosterically facilitates peptide bond formation” needs a reference.

Line 437- In Fig. S1 graph B, indicating the level of p-4EBP1/4EBP1, the point at 6 h has been misplaced.

Line 455- Puromycin incorporation assay.

Although I remained convinced of the result, the technical quality of this western blot image is not so high. Why only low MW proteins are visible? Which is the amount of cell lysates loaded on gels? Is there a reason for such a short time for Puromycin incorporation (5 min) and for the concentration of Puromycin (2 μ M would be a better choice)?

Can the authors comments on this?

Line 474 – eIF5A depletion by siRNA

It should be specified if the antibody used in the western blot is recognizing both isoforms of eIF5A. Antibodies specific for the two proteins are available and often the name eIF5A has been used to refer to eIF5A1.

Line 485 – Ribosome stalling assay

The authors should explain why the GC7 concentration has been raised to 40 μ M instead of 10 μ M used in all other experiments.

Line 494 – The authors state that “Inhibition of eIF5A activity result in reduced translation initiation through an eIF2 α -dependent feedback mechanism”. I believe that it is premature to state that. Data obtained so far do not really suggest that the mechanism is eIF2 α -dependent.

Line 551 – The decrease in total eIF5A in the presence of GC7 is not really consistent throughout the different experiments. Here (Fig.3A) it appears to be reduced already at 3h while in Fig 1A it is not and in Fig. 4A even at 24h it is not reduced. Do authors have an explanation for this?

Line 589 – OMA1 and DELE1 depletion with siRNA: the knockdown of the two mRNA has been shown by RT-PCR but a western blot showing the decrease of the two proteins should be presented as well.

Line 603 – The graphs in Fig 6A and 6B should be homogeneous: they should have the same Y axes and X axes should start at 0.

Line 711 – The authors interpret their results by proposing that “eIF5A inhibition could induce the down-regulation of proteins within the TIM23 complex”. This statement should be easily verified in the proteomic experiment. Is there any evidence of a down-regulation of any of the proteins belonging to the TIM23 complex? If not do authors have any explanation for this

The results of the SILAC proteomic experiments (the list of the proteins down-regulated in the presence of GC7) should be made available.

Version 1:

Reviewer comments:

Reviewer #1

(Remarks to the Author)

While the authors have addressed some of the initial points raised during the first round of review, the two core concerns remain inadequately resolved. As a result, the strength of the central conclusions continues to be questionable. My detailed comments are as follows:

1. Specificity of GC7 and demonstration of DHPS-dependence

The central claim of a time-dependent translational feedback mechanism mediated by EIF5A hypusination remains

insufficiently supported. I had previously suggested a time-course analysis using inducible (e.g., doxycycline-inducible) DHPS knockdown to convincingly demonstrate that the effects observed with GC7 are indeed attributable to DHPS inhibition. Instead, the authors have employed a constitutive shRNA approach targeting both DHPS and DOHH, observing effects on initiation but not elongation.

This is not equivalent to the requested approach and fails to capture the temporal dynamics shown with GC7. I strongly recommend repeating the experiments using an inducible DHPS knockdown system and collecting time-point data that parallel the early vs. late effects seen with GC7 treatment.

Additionally, I had proposed the use of EIF5A K50R mutants to test the role of hypusination directly. The authors claim that these mutants are unstable and cite Khang et al. (JBC 1994, Ref #34 in the manuscript and Ref #1 in the rebuttal letter) to support this. However, that study provides no data regarding K50 hypusination or EIF5A protein stability. In contrast, Sun et al. (J. Cell. Physiol. 2010; 223(3):798-809) demonstrate that WT and K50R EIF5A1 are expressed at similar steady-state levels (see their Figure 2). Therefore, the authors are still encouraged to test this mutant construct as it remains a valid and mechanistically informative tool.

2. In vivo relevance

No in vivo validation has been provided. As noted previously, animal models would be essential to assess both the pathophysiological significance and therapeutic relevance of the proposed mechanism. Without in vivo experimentation, the translational claims remain speculative.

3. EIF5A1 vs EIF5A2

The issue of isoform specificity has not been satisfactorily addressed. The authors assert that EIF5A1 is the predominant isoform in A549 cells and cite Elvira-Blázquez et al. (EMBO J 2024) in support. However, this reference does not provide isoform-specific expression data in A549.

Contrary to the authors' claim, other studies show that EIF5A2 is detectably expressed in A549 cells. For example, Cell Death Dis. (2022) 13:683 reports EIF5A2 protein expression in this cell line. Furthermore, EIF5A2 levels are upregulated following EIF5A1 knockdown, as shown in Cell Communication and Signaling (2023) 21:54 (see Figure 2). This suggests potential compensatory or redundant functions that have not been adequately explored.

Moreover, in Figure S3A, the EIF5A1/2 blot shows only a single band. Given the distinct molecular weights of the two isoforms, a doublet should be observed. The lack of isoform discrimination raises concerns about antibody specificity or gel resolution, and should be clarified.

In summary, the manuscript continues to suffer from key conceptual and experimental gaps:

- The reliance on GC7 without sufficient genetic confirmation of DHPS involvement;
 - The lack of in vivo data undermining translational claims;
 - The unresolved EIF5A1/EIF5A2 distinction, which raises questions about isoform-specific functions and compensation.
- Given these outstanding issues, I do not consider the manuscript suitable for publication in Nature Communications in its current form.

(Remarks on code availability)

Reviewer #2

(Remarks to the Author)

I thought that the authors addressed my comments in a satisfactory manner and I have no further concerns.

Sincerely

I/Topisirovic

(Remarks on code availability)

Reviewer #3

(Remarks to the Author)

The authors have made significant changes to the manuscript that address all my prior concerns.

(Remarks on code availability)

Reviewer #4

(Remarks to the Author)

The revised manuscript entitled "eIF5A-dependent feedback inhibition from mRNA translation elongation to initiation limits tumour cell proliferation" by Sfakianos et al. focuses on the activity of the translation factor eIF5A and proposes a new pathway of translation regulation.

The use of the well-known hypusination inhibitor, GC7, in cancer cell lines, helps to shed light on the molecular

mechanisms, showing that inhibition of eIF5 activity has effects not only on translation elongation but, with a feedback mechanism, also on translation initiation and that eIF5A effects do not depend on specific amino acids motifs. By using GC7 in combination with ISRIB the authors manage to uncouple the two processes and to identify the targets of the two pathways. The authors present data demonstrating that the link between eIF5A and translation initiation involves mitochondrial dysfunction, activation of mitochondrial proteases which in turn induce eIF2 phosphorylation by HRI and PERK kinases. In conclusion results suggest that GC7 could represent a potential pharmacological approach to target cancers which are particularly dependent on mitochondrial function for their proliferation.

The manuscript is very interesting and presents novel findings shedding light on the mechanism of action of an important translation regulator such as eIF5A and on its potential use as pharmacological target. The methods are detailed and have been implemented according to the revision

In my opinion the authors have replied to the revisions required and have offered new evidence to support their conclusion, therefore, the manuscript can be accepted for publication in this journal.

(Remarks on code availability)

Version 2:

Reviewer comments:

Reviewer #1

(Remarks to the Author)

I appreciate the effort made in revising the manuscript. However, the key concerns remain unresolved, particularly regarding the specificity of the experimental system and the robustness of the mechanistic conclusions.

The central claim — that inhibition of elongation through reduced eIF5A activity leads to feedback inhibition of translation initiation — is conceptually interesting. However, this model is based entirely on experiments using GC7, a compound with known and emerging off-target effects. In particular, data presented at the 2024 International Conference on the Biological Roles of Polyamines (Kobe, Japan) reported that GC7 can inhibit mitochondrial complex I, which is known to induce energetic stress, activate the integrated stress response (ISR) via eIF2 α phosphorylation, and lead to global inhibition of translation initiation. This mechanism closely mimics the effects described in the manuscript, raising the possibility that the observed phenotype is not mediated by inhibition of eIF5A hypusination, but rather by indirect activation of ISR due to mitochondrial dysfunction.

These concerns are reinforced by observations from our group, which has also noted off-target effects of GC7 in unrelated experimental contexts, consistent with the possibility that the observed translation phenotypes may not be solely due to inhibition of eIF5A hypusination.

This issue is particularly critical because the temporal sequence proposed by the authors — early inhibition of elongation (3 h), followed by delayed inhibition of initiation — is demonstrated exclusively using GC7. Since this biphasic response is a central conceptual point of the manuscript, the lack of orthogonal validation substantially limits the strength and specificity of the conclusions.

While the authors argue that genetic approaches may not fully recapitulate the timing of pharmacological inhibition, this does not justify their exclusion. At least two complementary strategies are both feasible and necessary:

An inducible DHPS knockdown or knockout system, enabling controlled, time-resolved hypusination inhibition while avoiding off-target drug effects.

A DHPS knockout cell line rescued with a non-hypusinable eIF5A mutant (e.g., K50A or K50R). Although the authors mention potential instability, this is not demonstrated experimentally. Titration of the expression vector could mitigate this issue. The immunoblot shown at 72 h is not informative in this regard and is not directly comparable to earlier time points (e.g., 48 h).

The possibility that protein stability and response dynamics (e.g., DHPS, eIF5A) vary across cell types is another relevant factor that is disregarded by the authors.

In summary, while the study raises interesting hypotheses about translation regulation, the exclusive reliance on GC7, without genetic validation, prevents a clear attribution of the observed effects to eIF5A hypusination. Additional experiments are required to disentangle direct effects from GC7-specific off-target stress responses.

(Remarks on code availability)

Reviewer #5

(Remarks to the Author)

I have carefully reviewed the revised manuscript by Sfakianos & Raven et al., along with the reviewer-1 comments and the author rebuttals. The central concern raised by Reviewer-1 relates to potential off-target effects of the GC7 compound. There is indeed some literature describing eIF5A-independent cellular effects of GC7 (e.g. on autophagy - Oliverio et al., 2014), and as with most small-molecule inhibitors, some degree of off-target activity is not unexpected (e.g. for instance, even some of our most specific kinase inhibitors used in clinic hit dozens of other targets). The critical question, however, is whether the molecular and cellular effects of GC7 reported here can be confidently attributed to inhibition of eIF5A.

To address this point, the authors present several independent lines of evidence:

1- RNAi-mediated depletion of eIF5A reproduces the effects of GC7.

2- A distinct small-molecule inhibitor of the pathway (CPX), which targets DOHH rather than DHPS (target of GC7), also recapitulates the GC7 effects.

3- RNAi-mediated depletion of DOHH and DHPS similarly phenocopies GC7 treatment.

4- The observed effects can be rescued by ISRIB, demonstrating that the downstream pathway identified here is essential for the cellular phenotype.

5- A rescue is also achieved by mutating eIF2 α on S51, the inhibitory phosphorylation site mediating the initiation block downstream of eIF5A.

It would be highly improbable for the experiments 1, 2, & 3 to yield concordant results solely due to off-target effects. As a result, these findings strongly support the conclusion that the cellular effects of GC7 are mediated via inhibition of eIF5A function. Furthermore, experiments 4 and 5 provide compelling evidence of functional epistasis between the the translation-initiation regulation described here and the phenotypic consequences of GC7 treatment (as opposed to other potential off-targets effects of this compound), further strengthening the conclusions of the study.

Taken together, the presented evidence in my view is very convincing. While I fully appreciate the initial concerns of the reviewer-1 regarding GC7 specificity, the breadth of additional supporting data, using multiple different means of targeting the same pathway, substantially mitigates this issue. I therefore recommend that the manuscript is accepted for publication. Importantly, I do not believe the inducible knockdown experiment that the reviewer is suggesting in their latest comments will add anything further, as RNAi still takes a while to significantly deplete a protein target (inducible or not), so the result of this experiment will be basically the same as what is already shown in terms of RNAi-mediated eIF5A depletion.

My only minor suggestion to the authors is that in the Discussion section they may want to acknowledge the potential for GC7 having off-target effects, but also clearly articulate why their experimental results argue strongly against this being a concern. Summarising the key supporting lines of evidence mentioned above would pre-empt similar concerns from future readers and would further strengthen the manuscript in my opinion.

(Remarks on code availability)

N/A

REVIEWER COMMENTS

Reviewer #1 (Remarks to the Author):

1) Approach: the work is almost entirely based on the use of the DHPS inhibitor GC7 to inhibit EIF5A function, mostly using one cell line (A549).

It is generally believed in the field, and documented in some reports (e.g. Oliverio et al. Amino Acids 2014 DOI 10.1007/s00726-014-1821-0; Landau et al., JBC 2010 <https://doi.org/10.1074/jbc.M110.106419>), that GC7 has effects unrelated to DHPS inhibition and may directly target mitochondrial proteins. Even if lower doses and specific cell lines have been used in this work, authors have not provided evidence of the specificity of GC7 under their conditions. In Figure S3 it is shown that the knockdown of both EIF5A1 and 2 induces markers of ISR and decrease of translation rate. However, the authors have knocked down EIF5As, but not DHPS, the actual target of GC7. To demonstrate the time-dependent effect of decreased/abrogated EIF5A hypusination, the authors should use inducible DHPS knockdown and/or cells stably expressing a mutant EIF5A lacking the hypusinated lysine (i.e. EIF5AK50R) in all the experiments shown in the manuscript. As the paper stands now, only the consequences of GC7 treatment are carefully documented, but this is not a trustable reflection of decreased EIF5A hypusination and function.

As the reviewer acknowledged, the concentrations used in studies which suggest that GC7 has possible DHPS-independent effects were in considerable excess over those used herein and in addition to much longer time points, increasing the possibility of assessing secondary and not primary events. To limit the potential of indirect effects we used a concentration **10-fold** and **20-fold** lower than Landau et al and Oliverio et al, respectively, and our mechanistic investigations were carried out over **30 minutes – 24 h** rather than 72 hr treatment (eg as in Landau and Oliverio).

Utilising eIF5A mutants is problematic; hypusination stabilises eIF5A¹, and mutant eIF5A-K50 proteins are generally unstable, which would make these experiments challenging. It is also likely that this approach would inhibit on translation prior to the addition of GC7 making it hard to draw conclusions. We have instead added the following data to provide more evidence for the specific effect of GC7 on eIF5A activity:

1. We have used siRNA depletion of both enzymes involved in hypusination (DHPS and DOHH) and observed that DHPS/DOHH depletion resulted in the induction of p-eIF2 α (Fig S3G+H), correlating with observed effects of eIF5A1/2 knockdown (Fig S3A-D) and treatment with GC7 (Fig 1).
2. We used a different inhibitor of the hypusination pathway, ciclopirox (inhibitor of DOHH), and observed the induction of p-eIF2 α /ATF4 (Fig S6B-D) and inhibition of protein synthesis (Fig S6J+K), correlating with GC7 treatment.
3. Ciclopirox also reduced mitochondrial protein expression (Fig S6E-I), again correlating with our observations with GC7.
4. We determine the mitochondrial function of both A549 and MCF10A cells. These data show that A549 have increased basal respiration rates and no reserve capacity (Fig 6D), suggesting that A549 cells may be more dependent on their mitochondria

for energy production and providing a further link to why they are sensitive to GC7. In support of this observation, basal respiration of A549 is reduced within 6 h GC7 treatment, in line with our proteomic analysis (Fig 2), whereas MCF10A were unaffected. These data provide additional mechanistic understanding of how GC7 is impacting A549 mitochondria function.

While we accept that these additional data are correlative, they strongly suggest that the effects observed of GC7 on A549 and mitochondria are specific to the inhibition of hypusination.

2)

a) The group of Chaim Kahana described the effect of EIF5A inhibition on translational initiation (Landau et al., JBC 2010 <https://doi.org/10.1074/jbc.M110.106419>). Using mouse fibroblasts, they observed that treatment with GC7 inhibited translational initiation and induced eIF2alpha phosphorylation, although with kinetics that were different from this submitted paper. Indeed, they saw an early (from 3h) increase of phosphorylated eIF2alpha (Fig 6B), in contrast with the present manuscript, where the authors claim a later (24h) activation of this protein. Of note, upon interference of EIF5A, Kahana and collaborators failed to reproduce the GC7-mediated decrease of initiation (Fig 4), whereas they noticed decreased elongation. This discrepancy was attributed to non-specific effect of GC7. Authors (who have also used mouse fibroblasts, Fig 1E) need to address these conflicting data.

In our original submission we normalised all western blot treated values to the untreated control, which provided a fold change value relative to 1 (to take into account any potential changes in the overall intensity of the blot i.e., due to antibody batch variability). However as noted by reviewer 3 (comment #3), this analysis does not allow for variation in the mean of the untreated sample, making valid statistical analysis challenging. Therefore, we have re-analysed all western blot data within the manuscript normalising each lane to the mean of all lanes on each blot. All densitometry has been updated and reanalysed using the appropriate statistical test, which is now indicated in each figure legend.

The new analysis does not change the narrative of the manuscript however there are some subtle differences in Figure 1. There is an elongation block at 3 hr which precedes p-eIF2 α , but this is now significantly increased from 6 hr (previously elevated from 6 hr but only significant at 24 hr) and similar to the observations by Landau et al. Moreover, significant elevation of p-eIF2 α now correlates with the significant increase in ATF4 expression, as would be expected. Our updated polysome profile analysis (Fig 1I) also shows an increase in free 80S from 6 hr, correlating with the inhibition of initiation.

It is challenging to make direct comparisons between Landau et al. and our study due to important differences in experimental design. As discussed above, Landau et al. used a high GC7 concentration (100 μ M) and treated for up to 72 hours, whereas we used a lower concentration (10 μ M) and our mechanistic investigations were carried out over 30 minutes – 24 h. Indeed, the siEIF5A data (Fig 4) referred to by the reviewer was a 6-day timepoint and it is unclear what secondary or tertiary effects of eIF5A depletion (i.e., translational reprogramming and cellular adaptation) may have occurred in that time. Moreover, Landau

et al. did not determine if the inhibition of elongation preceded the inhibition of initiation in their model, and did not determine cell viability in any of their assays, which could impact the interpretation of their data. We have now acknowledged this study in the discussion (page 27, line 1).

In another paper (Sun et al., *J Cell Physiol.* 2010 Jun;223(3):798-809.doi: 10.1002/jcp.22100) it was shown that deletion of EIF5A prevents apoptosis, while its overexpression induces cell death, causing activation of the intrinsic mitochondrial pathway. Therefore, the observations are in stark contrast with the submitted report, where EIF5A inhibition with GC7 induces apoptosis. Of importance, Sun et al. showed that the cytotoxic effect of EIF5A was not due to hypusination since overexpression of hypusination-defective EIF5A mutants still induced apoptosis (Fig 2 and 3). This conflicting data should also be carefully addressed and taken into account when planning the experiments.

Some of our original flow cytometry cell death data was confounded by an issue with the cytometer identified after submission and experiments have been repeated in biological triplicate. These data show that GC7 does not induce cell death (Fig S9A), correlating with Sun et al, but rather leads to an accumulation of cells in the G1 phase of the cell cycle (Fig S9B-C) and reduces cancer cell proliferation (Fig 6A-B). Importantly, we have confirmed that the reduction of cell proliferation is specific to the phosphorylation of eIF2 α , as it is reversed by ISRIB (Fig 6G), and may lie downstream of mitochondrial ISR signalling.

3) Clinical relevance: Another major limitation of this study is the lack of in vivo evidence. A central claim of this paper is that hypusinated EIF5A inhibition may represent a plausible novel strategy to target tumors dependent on mitochondrial function. However, only in vitro studies are shown (Fig 6). It would be important to provide solid evidence of the pathophysiological importance of these findings using relevant in vivo models of tumors harboring inducible DHPS and/or EIF5AK50R mutants to recapitulate the in vitro observations.

We have expanded our in vitro studies to include the analysis of mitochondrial function of A549 and MCF10A cells (Figure 6D-E and S10C-G). Taken together with the original ATP assay data, these suggest that A549 are highly dependent on their mitochondria in the steady state and have no reserve mitochondria capacity. Conversely MCF10A have less basal energy demands and possess a large reserve capacity of mitochondrial function. Therefore, the lack of reserve capacity or reliance on mitochondria may correlate with sensitivity to GC7. In support of this hypothesis, we now show that four different tumour derived malignant pleural mesothelioma (MpM) cell lines, which are reliant on OXPHOS², are sensitive to GC7 (Fig S11A-D). Moreover, HeLa cells have minimal mitochondrial reserve capacity^{3,4} and are sensitive to GC7 (Fig S4A), whereas MCF7 and HT-29, which have larger reserve mitochondrial capacities⁵⁻⁸, are insensitive to GC7.

Taken together, these data suggest that the sensitivity to GC7 is related to mitochondrial function and provides the basis for future studies to explore the viability of targeting OXPHOS-dependent cancers with inhibitors such as GC7. However, as we were unable to conduct in vivo studies, we have toned down statements throughout the manuscript and include these as suggestions or hypothesise that could be followed up in future work.

Additional Points

1- The representative Figure 1A looks different from the quantifications shown on the side (B-F).

We ensured that the representative blots in Fig 1A were all from the same replicate, rather than choosing individual blots across the 7 replicates, as this was more objective. In addition, as detailed in response to major comment #2, we have reanalysed all our western blot densitometry data (Fig 1B-F) and the new analysis more accurately reflects the representative blots shown in Fig1A. We have also included the individual datapoints on each graph to illustrate the variation observed over 7 biological replicates.

2- The decrease of hypusination (1A-B) after 0.5-3-6 hrs is quite modest (20-25%). It is difficult to concur that it might be biologically relevant.

We appreciate that the decrease in hypusination is modest at the early timepoints, however, it is significant from 3 hr and correlates with a slowing of translation elongation (shown by an increase in heavy polysomes in figure 1J – orange trace), which suggests GC7 is functionally active and the impact on hypusination is biologically relevant in our cell culture model.

3- The downregulation of total EIF5A after longer GC7 exposure (1A-C) is also concerning and not well explained.

We apologise that this was not clearly explained in the original submission. eIF5A becomes unstable when it is not hypusinated¹ and inhibition of hypusination has been linked to reduced eIF5A stability⁹. Therefore, the downregulation of eIF5A observed at later timepoints is likely due to less stable non-hypusinated eIF5A.

We have added a clearer explanation of this phenomena with the appropriate reference to the results section (page 17, line 10-12).

4- The claimed temporal changes of p-eEF2 and p-eIF2alpha and ATF4 are not very convincing: judging from figures 1A, E, F, variable and subtle differences of questionable relevance seem to occur in the earlier time points.

Our revised western blot analysis shows inhibition of elongation (shown by significant increase in p-eEF2 at 3 hr) precedes the inhibition of translation initiation (shown by the significant increase of p-eIF2 α at 6 hr). Furthermore, the increase in p-eIF2 α correlates with the increase in ATF4 expression. As this was a time-dependent correlation of different signalling events, we proceeded to determine the relevance of p-eIF2 α within this mechanism using MEF S51A that contain unphosphorylatable eIF2 α (Fig S1E). These data showed a loss of ATF4 expression following GC7 treatment and suggested that the phosphorylation eIF2 α during the early response is relevant to the entire mechanism.

We have added additional text to better explain our observations that we trust are clearer. We have also added the caveat that we are observing a correlation between these events (page 17-18).

5- Data in Fig S1 are also not convincing. The early increase of puromycin incorporation in Fig 1G does not look very strong.

We have repeated all puromycin incorporation assays using an optimised protocol to visualise larger polypeptides (in line with reviewer 4, comment #4). These data show that puromycin incorporation is significantly increased at 0.5 h (Fig1 G and 1H).

FigS1 A-D were in parallel to the data in Fig1A (7 independent biological replicates) which suggest the initial response to GC7 from p-eEF2 through to p-eIF2alpha/ATF4 is independent of mTOR and 4EBP regulation. We subsequently utilised S51A MEFs to confirm that GC7-induced ATF4 expression was mediated specifically through p-eIF2alpha (E-J).

6- Figure S3: only one band is shown for EIF5A1 and 2. These proteins migrate differently and typically form a doublet. Do A549 cells express both isoforms? In this case, both bands should be detected.

We apologise that we did not make this clear in the text. The antibody we have predominantly used is specific to eIF5A1 and does not detect eIF5A2. We used this antibody as eIF5A1 is the main isoform in A549¹⁰ and HeLa⁹ cells, and both cell lines express very low levels of eIF5A2¹¹. We have added a clarification of the expression in A549 to the results section (Page 17, line 10-12) to make clear we are only observing eIF5A1 levels in our data.

We have repeated eIF5A1/2 depletion and probed using an antibody that recognises both isoforms (Fig S3A-D). We have also amended the naming of all eIF5A blots depending on whether the antibody used recognised only eIF5A1 or both eIF5A1/2 isoforms.

7- Figure S4A why 40 uM of GC7 was used in HeLa cells and 10 uM in A549? Did the authors perform dose-response studies?

We originally included a titration of GC7 in HeLa cells (now in Fig S4A). Although growth was significantly inhibited using 10 μ M GC7 from 48 hours, detection of ribosome collisions is challenging, so 40 μ M was used as this dose robustly inhibited cell proliferation, and we were attempting to induce as many stalled ribosomes as possible. We have added text to the results section to clarify this decision (page 19, line 3).

8- Figure 6A-B the effect of GC7 on MCF10A is defined as non-significant but the proliferation rate shown in Figure 6B seems reduced by GC7. Perhaps the number of replicates should be increased.

Cell index (output from the RTCA instrument) is a measure of cell adherence that we infer as cell number and proliferation. The original data included a normalised cell index value and this has been replaced with raw cell index values from a representative experiment (Fig 6B) and statistical analysis at 48 h from 3 biological replicates (Fig 6C). We subsequently validated changes in cell index with parallel experiments to determine cell cycle state and cell viability (Fig S9A-C). As these data suggest no cell cycle arrest or cell death is observed in MCF10A cells, unlike in A549 cells, it is highly likely that GC7 does not have a significant effect on MCF10A cell proliferation and further replicates beyond the 3 independent biological experiments already included would affect this.

Reviewer #2 (Remarks to the Author):

Major comments:

1-Throughout the manuscript, it remained rather unclear how densitometry was performed (e.g., what was used to normalize the data). Considering that the article heavily relies on this analysis, this information should be provided in each figure legend.

We apologise that this was not clear in the original submission. The signal for phosphorylated proteins is normalised to the corresponding total protein level. Signal for non-phosphorylated proteins is normalised to Actin, except hypusination which was normalised to eIF5A1. We have now added this information to the methods (page 7).

After normalising each band to the appropriate loading control, we originally normalised all treated values to the untreated control, which provided a fold change value relative to 1 (to take into account putative changes in the overall intensity of the blot - i.e., due to antibody batch variability). However as noted by reviewer 3 (comment #3), this analysis does not allow for variation in the mean of the untreated sample, making valid statistical analysis challenging. Therefore, we have re-analysed all western blot data within the manuscript to normalise each band intensity to the mean intensity across all blot lanes. All densitometry has been updated and reanalysed using the appropriate statistical test, which is now indicated in each figure legend.

The new analysis does not change the narrative of the manuscript, the effect of ISRIB (Fig 3), the eIF2 α activated by GC7 (HRI and PERK – Fig 4, S8) or the activation of DELE1 and OMA1 (Fig 5).

The new analysis has resulted in some small changes to Fig 1 A-F. For example, p-eEF2 is not significantly increased until 3 hr (previously 0.5 hr), but the elongation block at 3 hr still precedes p-eIF2 α , which is now significantly increased from 6 hr (previously 24 hr). Moreover, elevation of p-eIF2 α now correlates with a significant increase in ATF4 expression, as would be expected. Our analysis of 4EBP1 phosphorylation now shows a significant decrease from 18 hr (rather than 24 hr), however this event is not present during the early response to GC7.

2-Line 455: “Western blot analysis of puromycin incorporation revealed an augmented incorporation in the early stages of GC7 treatments, which is likely indicative of an accumulation of slow-moving ribosomes on the mRNAs and a reduction in elongation”. I am a bit confused by this statement, as it is somewhat unclear to me how slowed elongation may result in the induction of puromycin incorporation. Can the authors elaborate on this? These effects appear to parallel the induction of ISR and thus the alternative explanation may be that the shut-down of global protein synthesis may at least in part occur via induction of ISR, while the effects of modest inhibition of eIF5A hypusination (early time points) on global mRNA translation may be marginal. Although authors speculate that relatively late downregulation of global protein synthesis upon GC7 treatment may be due to inhibition of initiation, it appears warranted to test the effects of GC7 treatment on

global protein synthesis in S51A eIF2 α KI MEFs. Finally, the technical quality of the figure 1G is not the greatest and should likely be improved.

We apologise that this result was unclear. Puromycin will incorporate into the A-sites of ribosomes and if we inhibit elongation but maintain initiation, there would be an initial increase in the number of ribosomes on an mRNA until translation initiation is inhibited. Therefore, we hypothesise that the observed increase in puromycin at 0.5 hr is unlikely to be increased mRNA translation, but rather an increase in slow moving ribosomes accumulating on mRNAs. We have amended the text in the results section to make this clearer (page 18, line 4-7). We have now also repeated our puromycin assays using an optimised protocol which is now included in Fig 1G and H.

Our densitometry reanalysis now suggests the inhibition of translation initiation by p-eIF2 α is significantly elevated from 6 hr, suggesting that activation of the ISR is likely responsible for global protein synthesis inhibition, which is followed by a decrease in mTOR signalling and dephosphorylation of 4EBPs. We have added an explanation of these data to the results section (page 17, line 15-19).

3-Polysome profiles presented in figure 1I may not be of sufficient resolution to substantiate the authors claim that 80S peak remains unchanged upon GC7 treatment. Namely, in the provided polysome tracings 80S peak appears to be unidentifiable.

We have replaced these polysome profiles with ones of increased resolution and included them within the same plot to highlight the subtle differences observed. There is a small but reproducible increase in heavy polysomes after 0.5 hr treatment, suggesting accumulation of slow-moving ribosomes on mRNA. From 6 hr, there is an increase in free 80S coupled with a small decrease of heavy polysomes, indicating inhibition of initiation, which correlates with the significant increase of p-eIF2 α in the new analysis of western blot data (expanded in response to point 1). From 24 hr, there is a loss of polysomes and increase in free 80S, suggesting robust inhibition of translation initiation.

4-Quantification in S3H does not seem to match blot in S3G.

We have repeated the puromycin incorporation assay following eIF5A1+2 knockdown using an optimised puromycin pulse protocol. We now include a representative western blot (Fig S3E) and quantification of 3 biological replicates (Fig S3F).

5-The relationship between tested proteins in Figure 3 and proteome analysis should be specified. Of note, it was hard to find out actual proteins that were affected in SILAC study as well as raw data associated with it. Were OMA1 and DELE1 levels affected by GC7? Also, the $\frac{1}{2}$ lives of ribosomal proteins are notoriously long and thus the authors may want to check what they may be for rpS6 and rpS25 as this may mask the effects on de novo synthesis of these proteins.

We apologise that the supplementary table for the SILAC proteomics analysis was omitted by mistake. We have now included the supplementary table (Table S5) to show all proteins changing with the corresponding fold change and p-value at each time point. We have also clarified the specific effect on the proteins shown in Figure 3 in the results section (Page 21, line 20-23).

One of the limitations of our SILAC proteomics approach is that we only detect newly synthesised proteins and lack the detection of less abundant proteins. Perhaps for these reasons we were unable to detect OMA1 and DELE1.

The half-life of RPS6 has been estimated ~6 hr in A549 ¹², so it is unlikely this will mask the effects of de novo synthesis in repose to GC7. Conversely, S25 has been estimated to have a half-life of between ~2.5 hr ¹³ and ~69 hr ¹², making the interpretation of S25 more challenging. We have therefore acknowledged this as a caveat when interpreting our data (page 28 , Line 5-9).

6- A549 and MCF10A cells are highly genetically heterogenous. Thus, it was thought that it would be warranted to **include isogenic non-transformed and transformed cell lines** in order to determine whether the effects of GC7 are specific to cancer cells and/or tone down associated claim.

In line with a comment from reviewer 1 (Major pint #3 Clinical relevance), we have toned down all claims that the effects observed are dependent on cancer or non-cancer cells. Our comparison of A549 and MCF10A now only focuses on the differences in basal respiration rates and mitochondrial function, suggesting cell lines more reliant on OXPHOS, or those with limited reserve mitochondrial capacity, may be more sensitive to GC7 (see response to comment 7 below). To test this hypothesis, we have now included growth assays for malignant plural mesothelioma (MpM) cell lines, which are more reliant on OXPHOS (Fig S11). Consistent with our hypothesis, these data suggest the MpM is also sensitive to GC7, whereas other cell lines that have been reported to be less dependent on OXPHOS or have higher reserve OXPHOS capacity, such as MCF7 and HT-29 ⁵⁻⁸, are less sensitive to GC7 (Fig S4A).

7-It appears somewhat ambiguous to define A549 cells as those having “high OXPHOS” levels. What was the criteria for this? When A549 cells were cultured in galactose containing media, the ATP production was decreased (Figures 6F and G) which suggests that their mitochondria are actually dysfunctional, and unlikely to exhibit “high OXPHOS” levels. Moreover, when A549 cells were forced to rely on mitochondrial metabolism by culturing them on galactose without glucose exhibited enhanced sensitivity to GC7 (Figure 6H). Considering this, it was thought that the authors may want to reconsider whether the sensitivity to GC7 is linked to “high OXPHOS levels” or perhaps on mitochondrial dysfunction which appears to be more likely from the aforementioned experiments. Finally, it appears warranted to include some respirometry studies and determine bioenergetic profiles of investigated cell lines as well as the effects of GC7 on bioenergetics +/- ISRIB.

We have now carried out respirometry studies using the seahorse XF analyzer to compare oxygen consumption rates of A549 and MCF10A cells. These data indicate that A549 have increased basal respiration rates and no reserve capacity (Fig 6D), suggesting that A549 mitochondria are already functioning at maximum capacity. Upon GC7 treatment, the basal OCR of A549 is reduced within 6 h, whereas MCF10A were unaffected, which perhaps reflects their differences in basal respiration. Crucially ECAR indicates that A549 readily switch to using glycolysis (Fig S10G) in glucose conditions to ensure ATP production is maintained (Fig S10A) even though mitochondrial function is inhibited (Fig 6E, S10C).

However, when A549 are grown in galactose and forced to use their mitochondria for ATP production, ATP is reduced which suggests GC7 treatment may lead to mitochondrial dysfunction.

We have amended our conclusions to focus on the sensitivity to GC7 being related to reliance on mitochondria and/or lack of reserve mitochondrial capacity.

Minor issues:

1-More comprehensive introduction of ISR may be warranted.

We have expanded the introduction of the ISR in the results section prior to the use of ISRIB (page 21, line 5-11).

2-The article needs some polishing when it comes to consistency and editing. In addition, the figure legends were found to be somewhat incomplete whereby some important details (e.g., representation of the error bars i.e., SD or SE), number of replicates in some panels are missing, etc.) appear to be missing.

We apologise for the inconsistencies included in the original submission. We have now standardised the nomenclature used throughout the manuscript and standardised formatting across all figures (including axis labels, font, text size etc). In addition, we have amended all figure legends to ensure they include all the information required to understand the figure e.g., technique used, cell lines used, biological replicate number, defining error bars, detailed information of the statistical test used and p-values.

3-Alternative genetic approaches (e.g., depletion of DHPS) to parallel GC7 studies may be more appropriate than depleting eIF5A1/2.

We have now included siRNA depletion of both enzymes involved in hypusination (DHPS and DOHH) in parallel with eIF5A1/2 depletion and observe the induction of p-eIF2 α (Fig S3G+H). In addition, we have used a different inhibitor of hypusination (DOHH inhibitor - ciclopirox) which also induces p-eIF2 α , ATF4 expression, inhibits protein synthesis and reduces mitochondrial protein expression (Fig S3G-H and Fig S6B-K). Taken together, these data indicate that the effects observed in response to GC7 are specific to the inhibition of hypusination.

4-Although the immunogen used to generate eIF5A Ab appears to be proprietary, the authors should specify whether this antibody recognizes both eIF5A1 and 2. Alternatively, RT-qPCR should be used to validate the level of KD of each isoform by RNAi.

We apologise that we did not make this clearer in the text. The antibody we have used is specific to eIF5A1 and will not detect eIF5A2. We used this antibody as eIF5A1 is the main isoform in A549¹⁰ and HeLa⁹ cells, and both cell lines express very low levels of eIF5A2¹¹. We have added a clarification of the expression in A549 to the results section (Page 17, line 10-12) to make clear we are only observing eIF5A1 levels in our data. We have also amended the naming of all eIF5A blots to be eIF5A1 across all figures.

We have repeated eIF5A1/2 depletion experiments and probed using an antibody that recognises both isoforms (Fig S3A-D). We have also amended the naming of all eIF5A blots depending on whether the antibody used recognised only eIF5A1 or both eIF5A1/2 isoforms.

5-Some explanation should be provided why the cells were pretreated with ISRIB, and why ISRIB was not applied after GC7 if the inhibition of eIF5A hypusination precedes ISR. We have used ISRIB as tool to dissect the proteins that were reduced due to eIF5A inhibition (elongation effect) or activation of the ISR (initiation effect). Although inhibition of eIF5A activity (0.5 – 3 hr) does precede significant activation of the ISR (6 hr), there is a small, non-significant, increase in p-eIF2 α as early as 3 hr. We therefore pre-treated with ISRIB to ensure the inhibition of eIF2B would be fully alleviated prior to the addition of GC7. We have added this reasoning to the results section (page 21, line 16-18).

6-The role of mTOR in the observed phenomena remains unclear. This appears to be pertinent as GC7 appears to affect mTOR targets implicated in elongation (e.g., eEF2K, albeit this may be AMPK too) and initiation (e.g., 4E-BPs). Albeit it is understandable that this may be out of the scope of this study, this should be reflected in discussion section. In our initial experiments the phosphorylation of 4EBP1 did not decrease until 18 hr GC7 treatment (Figure S1), suggesting that in inhibition of translation initiation from 6 hr GC7 treatment was likely independent of mTOR and 4EBPs (Fig S1).

We agree that the role of mTOR/AMPK regulation of eEF2K is an important question, and it is currently being investigated in the laboratory. We feel it is beyond the scope of the current manuscript, however we have acknowledged the potential role of mTOR in the discussion (page 27, line 10-12).

7-Out of curiosity:

Are the levels of DOHH sensitive to mTOR inhibition and/or are they known to be regulated by ATF4?

Published data suggests that the translation of DOHH mRNA was not affected by Torin 1 treatment for 2 hr¹⁴. The level of DOHH mRNA was also unaffected by 24 hr Torin1 treatment and it was deemed not to be regulated by ATF4¹⁵.

I hope that the authors will find these comments helpful and of sufficient pathos.

Sincerely

I/Topisirovic

Reviewer #3 (Remarks to the Author):

Overall the work identifies and characterizes a feedback loop between the elongation and initiation phases of protein synthesis and shows that this relies in large part on signalling

from mitochondria to HRI and eIF2 phosphorylation. The western blots make it clear that other signals are also being relayed from eIF5A, eg via 4E-BP1 and eEF2 that are not further explored here. The work utilizes a wide range of molecular techniques and appears generally carefully performed and interpreted. The main conclusion is that cells relying more heavily on mitochondria including the cancer lines studied here are more susceptible to GC7 than other cells and that it may therefore have some therapeutic value that would require further study to assess. However what the key features of GC7-resistant cells verses sensitive cells is not well explored.

However I have quite a few questions and suggestions regarding the work and its presentation that are outlined below.

Points.

Major

1. The text states that MCF10A were identified as GC7 resistant cells but does not indicate what screen was done.

We apologise that text was unclear. MCF10A cells were not included in the crystal violet dose response (Fig S4A) and we instead carried out real time monitoring of proliferation with xCELLigence (Figure 6B) which suggested MCF10A were less sensitive to GC7. We have amended the text to indicate that the observation of MCF10A being less sensitive to GC7 comes from Figure 6B (page 24, line 11-13).

2. The methods make it clear different media were used to culture A549 and MCF10A cells. Could this change contribute to the observed differential GC7 sensitivity?

We think it is unlikely that the media used contributes to sensitivity to GC7. For example, although A549 were GC7-sensitive, MCF7 and HT-29 were not, even though all three cell lines were cultured in the same media (Figure S4A).

3. Western blot quantification. Throughout the MS almost every western it appears that 'lane 1' has been normalized to '1' for all signal quantification. The impact of this is that in the control samples there is no variation around the mean, while there is for all other samples. Thus for the statistics the samples all have unequal variance with the control. Instead of selecting 1 lane to normalize to it is a better practice to normalize all lanes to the mean signal across all the blot lanes. In that way all samples will have meaningful error bars and if there is no variation all lanes will be approximately 1. This permits the stats to be done in a valid way. Implementing this change will impact almost every main figure and many of the supplementary figures.

We appreciate the impact our original normalisation had on the statistical analysis. We have now re-normalised all individual bands to the mean signal across all blot lanes, as suggested. This has been applied to every western blot within the manuscript and every chart has been updated. Additionally, we have added an explanation of how densitometry and normalisation was carried out to the methods (page 7) and included details of the statistical tests used in each figure legend.

The new analysis does not change: the narrative of the manuscript, the effect of ISRIB (Fig 3), the eIF2k activated by GC7 (HRI and PERK – Fig 4, S8) or the activation of DELE1 and OMA1 (Fig 5D-I). However, the new analysis has improved the data presented in Fig 1 A-F

and there are some small differences from the original submission. p-eEF2 is not significantly increased until 3 hr (previously 0.5 hr), but the elongation block at 3 hr still precedes p-eIF2 α , which is now significantly increased from 6 hr (previously 24 hr). Moreover, elevation of p-eIF2 α now correlates with a significant increase in ATF4 expression, as would be expected. Our analysis of 4EBP1 phosphorylation now shows a significant decrease from 18 hr (rather than 24 hr), however this event is still not present during the early response to GC7.

Overall, this reanalysis of all western data has increased the robustness and statistical significance of our data.

4. P12 ribosome run off assay. From the data presented it is not obvious how differences in polysome density were calculated. It is clear from Supp Fig 2 that there is a large increase in free 80S upon HH treatment in both control and GC7 cells, but as the 80S peak goes dramatically off scale it becomes difficult to accurately quantify P/M ratios and rule out differential loadings. The methods description does not help the reader here and the presented polysome trace data do not appear different \pm GC7. In panel C should the timescale be in minutes? How the numbers in Fig 1J are derived is also unclear.

We apologise that our explanation of our methods was unclear. We normalised gradient sample loading by total protein (calculated by BCA assay immediately prior to running gradients), as we have done in all polysome analysis. We then quantified the area under the curve of the polysomes only (as previously described^{16, 17}) and did not use a P/M ratio. Changes in polysome density from 3 – 5 min harringtonine in DMSO and GC7 conditions were then determined relative to non-harringtonine treatment. We have amended the methods section accordingly to make this clearer (page 11).

In hindsight the analysis of our original data (linear regression to attempt to infer ribosome transit time, Fig1J) was over complicated and unnecessary. We have therefore replaced figure 1J with the area of polysomes after GC7 treatment and 5-minute harringtonine, which shows a significant increase in ribosomes remaining on the mRNA after GC7 treatment.

We hope this provides a clearer understanding of the methods employed and the data obtained.

5. Line 497. The 'dynamic' SILAC assay could be introduced better as many will not be as familiar with this type of assay. I think using the term 'nascent' to describe heavy labelled proteins that may have been made at anything up to 24hours before the assay point is misleading. In protein synthesis, 'nascent proteins' are typically defined as those that are undergoing synthesis and so are within/exiting the ribosome exit tunnel, not those made hours before. Please consider a different term. What cells were used for this assay? How many replicates are used, the Figure 2A appears to show 2 reps only for GC7 treated cells and 3 for the control, is this correct? If only 2 this will affect the statistical power. Have PCA or other whole dataset comparisons been done to visualise the data and analyze its consistency as a whole?

- We have added a sentence at the beginning of second results section to clarify the labelling media changes required for the technique (page 19).

- We appreciate the confusion with our previous terminology and have now replaced ‘nascent proteins’ with “newly synthesised proteins” throughout the manuscript and figure legends.
- These experiments were carried out in A549 cells using 3 independent biological replicates. The legend details were lacking and have now been updated to include this important information.
- We have added our PCA comparison of all samples and replicates to Supplementary Figure 5A to show the consistency of our data between replicates.

6. The analysis presented in Supplementary Figure 5D is not at all clear. What is being shown here? What do 0, 1, 2,3 denote on the X axis of each graph? What does AAs: 1-3 mean? We apologise that this was not described clearly in the figure legend. ‘AAs 1-3’ corresponds to the number of amino acids in the sequence. The X axis corresponds the number of long side-chain positively charged amino acids (K or R) in that sequence. We have now added this information to the figure legend.

7. What is the source of the ribosome pausing data in Figure s5G?

We have added this reference to the figure legend.

8. While the SILAC did not find any evidence of poly-proline involvement the authors should acknowledge that they have not sampled the entire proteome. Many poly-proline proteins are poorly expressed, others are secreted, so may not have been captured here. Hence the assay may not have been sensitive enough to capture many poly-proline substrates. Was any western blotting done of well-known poly-proline proteins not covered by the MS analyses?

No validating western blots of well characterised poly-proline proteins was carried out. We agree with the reviewer that we have not sampled the entire proteome and a limitation of this experiment is the sensitivity of detecting secreted or poorly expressed proteins. We have therefore added a caveat to the interpretation of data set to the discussion (page 27, line 16-17), which we hope the reviewer finds satisfactory.

9. Figure 3 The impact of ISRIB is quite modest, at best ~1.5 fold. It would be good to acknowledge the other impacts seen eg on 4E-BP phosphorylation suggest involvement of mTOR and eg eEF2K indicate wider impacts.

Although the impact of ISRIB was modest, we were able to observe a significant rescue in expression of RPS6, RPS25 and TOMM20, suggesting that the reduction of protein expression was dependent on the activation of the ISR.

In our initial experiments the phosphorylation of 4EBP1 did not decrease until 18 hr GC7 treatment (Figure S1), suggesting that the inhibition of translation initiation from 6 hr GC7 treatment was likely independent of mTOR and 4EBPs (Fig S1). The role of mTOR and AMPK regulation of eEF2K is an important question but we feel it is beyond the scope of the current manuscript and it is currently the subject of further investigation in the laboratory. We have instead acknowledged the potential role of mTOR in the discussion (page 27, line 8-12).

10. Figure 4/Supp Fig 7. siRNA experiments. what cell type used here?

A549 cells were used for these experiments. We have now added this information to both figure legends.

11. The HRI activation model proposed invokes processing and activation of DELE1 by OMA1. Has a change in isoform L-DELE1 to S-DELE1 been observed in MS data with GC7 or via western blotting? This would further support the proposed model.

One of the limitations of our SILAC proteomics approach is that we only detect newly synthesised proteins and lack the detection of less abundant proteins. Perhaps for these reasons we were unable to determine DELE1 cleavage. We attempted to monitor DELE1 cleavage by western blotting, however this was not possible due to the poor specificity of commercially available antibodies ¹⁸.

12. Do GC7 resistant MCF10A cells exhibit the mitochondrial morphology changes seen for A549 cells following GC7 treatment (ie as shown in Figure 5A/B for A549s)

The comparison of mitochondria of both cell lines is an important question. To fully answer this question, we now include respirometry studies to compare A549 and MCF10A mitochondrial function and oxygen consumption rates (Fig 6D), including following treatment with GC7 (Fig 6E and S10C-G). These data show that MCF10A have a large reserve mitochondrial capacity, whereas A549 have no reserve capacity, suggesting that A549 may be more reliant on their mitochondria to produce the ATP required for their increases metabolic rates (Fig 6D). Moreover, whereas treatment with GC7 severely reduces OCR of A549 cells, MCF10A are generally unaffected, suggesting that their lower levels of basal respiration and possibly more functional/healthy mitochondria may enable them to tolerate more mitochondrial stress induced by GC7.

minor

13. Supplementary Fig 1B the graph X axis is out of order 18 hours before 6? Why?

This was a formatting error and has now been amended.

14. Supplementary Fig 1E/H. Was it known previously that p-eEF2 levels are very low in S51A MEFs?

Although not explicitly stated, published data suggests that p-eEF2 levels are low in S51A MEFs ¹⁹, however it is challenging to fully compare to WT MEFs.

15. Any thoughts as to why eEF2-P signalling is highly responsive to GC7 here?

We hypothesise that this may be due to the basal p-eEF2 levels being so low in these cells, and as eEF2 is a very abundant protein, this could lead to a large increase upon stress. In addition, eEF2K signalling has been shown to be elevated in S51A MEFs in response to oxidative stress ¹⁹. For this reason, we have refrained from making comparisons regarding p-eEF2 in the MEF experiments and instead focused on using S51A MEFs as a tool to determine that GC7-induced ATF4 expression was dependent on p-eIF2a.

16. P20 the reason for the switch to HeLa cells is not made clear.

Although HeLa cell growth was significantly inhibited using 10 μ M GC7 from 48 hours (Fig S4), detection of ribosome collisions is challenging, so we used the highest tested concentration that inhibited HeLa cell proliferation (Fig S4A) as we reasoned that would

have the most chance of causing detectable ribosome collisions. We have added this rationale to the results section (page 19, line 3-7).

17. Line 533. There are now 6 eIF2 kinases known including MARK2 and FAM69C not studied here.

We have not determined the role of either MARK2 or FAM69C within our model. To take this into account we have referenced both studies and added this as a caveat to the interpretation of our data in the discussion (page 27, line 6-8).

18. Line 534 eIF2AK2 is wrong here.

This has been amended.

19. Supplementary Fig 11. Adding panel numbers and labeling across this figure would help. The stained image text and graph labels are all too small.

We have added the appropriate labels and enlarged the figures.

20. Line 709 there is no reference 51.

We have amended this formatting issue.

21. Line 713. In the referenced study the polyproline stretch in TIM50 is implicated in eIF5A sensitivity, however while this is found in yeast TIM50 it is not conserved in human TIMM50, so the mechanisms are likely distinct. This could be clarified in the text.

We have acknowledged this important distinction (page 28, lines 26-28).

Reviewer #4 (Remarks to the Author):

The manuscript entitled “eIF5A-dependent feedback inhibition from mRNA translation elongation to initiation induces apoptosis in tumour cells” by Sfakianos et al. is focused on the activity of the translation factor eIF5A and proposes a new pathway of translation regulation.

The authors investigate the use of a well known hypusination inhibitor, GC7, in several cancer cell lines to better understand the molecular mechanism through which this molecule inhibits translation.

Results clearly show that inhibition of eIF5 activity by GC7, but also by knock down with siRNA, has effects not only on translation elongation but also on translation initiation and that the link between eIF5A and translation initiation involves mitochondrial dysfunction, activation of mitochondrial proteases which in turn induce eIF2 phosphorylation by HRI and PERK kinases. The authors claim that this pathway induces apoptosis in some cancer cell types which appear to be eIF5A sensitive and, therefore, suggest that GC7 could represent a potential pharmacological approach to target cancers which are particularly dependent on mitochondrial function for their proliferation.

The manuscript is interesting and presents novel findings shedding light on the mechanism of action of an important translation regulator such as eIF5A. It is well written and well documented in relation to the literature.

In my opinion it deserves publication in this journal with the following revisions:

Line 57: Acronyms (APC) must be specified.

We have specified this acronym.

Line 69: The statement “This positioning allosterically facilitates peptide bond formation” needs a reference.

We have added a reference for this statement.

Line 437- In Fig. S1 graph B, indicating the level of p-4EBP1/4EBP1, the point at 6 h has been misplaced.

We have amended this figure.

Line 455- Puromycin incorporation assay.

Although I remained convinced of the result, the **technical quality** of this western blot image is not so high. Why only low MW proteins are visible? Which is the amount of cell lysates loaded on gels? Is there a reason for such a short time for Puromycin incorporation (5 min) and for the concentration of Puromycin (2 μ M would be a better choice)?

Can the authors comments on this?

We have repeated the puromycin incorporation assays using an optimised protocol to visualise larger polypeptides (as suggested by the reviewer). Higher molecular weight proteins are now visible on the gel (Fig1G) and we have updated the methods section.

Line 474 – eIF5A depletion by siRNA

It should be specified if the antibody used in the western blot is recognizing both isoforms of eIF5A. Antibodies specific for the two proteins are available and often the name eIF5A has been used to refer to eIF5A1.

We apologise that we did not make this clearer in the text. The antibody we have used is specific to eIF5A1 and will not detect eIF5A2. We used this antibody as eIF5A1 is the main isoform in A549¹⁰ and HeLa⁹ cells, and both cell lines express very low levels of eIF5A2¹¹. We have added a clarification of the expression in A549 to the results section (Page 17, line 10-12) to make clear we are only observing eIF5A1 levels in our data. We have also amended the naming of all eIF5A blots to be eIF5A1 across all figures.

We have also repeated eIF5A1/2 depletion experiments and probed using an antibody that recognises both isoforms (Fig S3A-D). We have now amended the naming of all eIF5A blots depending on whether the antibody used recognised only eIF5A1 or both eIF5A1/2 isoforms.

Line 485 – Ribosome stalling assay

The authors should explain why the GC7 concentration has been raised to 40 μ M instead of 10 μ M used in all other experiments.

We used HeLa cells as the collision assay was previously optimised using this cell line. As HeLa cells were not as sensitive to GC7 as A549 (Fig S4), and the collision assay can be relatively insensitive, we used the highest tested concentration that inhibited HeLa cell proliferation (40 μ M) as we reasoned that would have the most chance of causing detectable ribosome collisions. We have added this rationale to the results section (page 19, line 4-6).

Line 494 – The authors state that “Inhibition of eIF5A activity result in reduced translation initiation through an eIF2 α -dependent feedback mechanism”. I believe that it is premature to state that. Data obtained so far do not really suggest that the mechanism is eIF2 α -dependent.

We appreciate that we are making correlations between our data. We have toned down our statement to make clear that it is “possibly in an eIF2 α -dependent manner”.

Line 551 – The decrease in total eIF5A in the presence of GC7 is not really consistent throughout the different experiments. Here (Fig.3A) it appears to be reduced already at 3h while in Fig 1A it is not and in Fig. 4A even at 24h it is not reduced. Do authors have an explanation for this?

Quantification of eIF5A1 from Fig 3 (below) suggests that the reduction at 3 hr was not significant over the 3 biological replicates, but the variation we observed is indicated by SD and the individual error points.

We generally observed a decrease in eIF5A levels following treatment with GC7 for 24 hr in A549 (Fig 1), HeLa (Fig S4C) and MCF10A (Fig S10C), which is in line with previous data suggesting that hypusination stabilises eIF5A¹. However, we experienced variability in eIF5A stability during some experiments (Fig 3, 4, 5, S2E and S8). The variability observed did not correlate with cell line batch, age or a change in batch of supplemented FBS. However, we used multiple different production batches of GC7 during the project and this may contribute to some of the variation observed. It is also possible that minor effects could be caused by small changes in cell confluence and drug uptake, which we have overcome by increasing replicates in some instances. However, as we were unsure of the exact effect on eIF5A levels, we have tried to avoid making many conclusions from these data, however, we can add any of this response to the discussion if the reviewers deem it necessary.

Line 589 – OMA1 and DELE1 depletion with siRNA: the knockdown of the two mRNA has been shown by RT-PCR but a western blot showing the decrease of the two proteins should be presented as well.

We have now added a representative western blot for OMA1 following knockdown (Fig S8C). However, the quality of commercially available DELE1 antibodies is poor and in support of other studies¹⁸, we were unable to find one that specifically recognised DELE1. As DELE1 protein has been shown to be rapidly degraded within 30 minutes^{20,21}, we reasoned that determining mRNA levels after 48 hr siRNA depletion (Fig S8A) should be sufficient. We have added this additional explanation to the results section (page 23, line 20-24).

Line 603 – The graphs in Fig 6A and 6B should be homogeneous: they should have the same Y axes and X axes should start at 0.

In line with a comment by reviewer 1, we have now replaced these plots with representative experiments containing raw cell index measurements (Fig 6A and B). We have added separate quantification of cell index from 3 biological replicates in both cell lines (Fig 6C).

Line 711 – The authors interpret their results by proposing that “eIF5A inhibition could induce the down-regulation of proteins within the TIM23 complex”. This statement should be easily verified in the proteomic experiment. Is there any evidence of a down-regulation of any of the proteins belonging to the TIM23 complex? If not do authors have any explanation for this

Our data does not identify all Tim23 complex members and is a limitation of our proteomics approach, however, Tim50, Tim44, HSPD1 and GrpE all decreased after treatment with GC7. We have added a comment on this and a reference to the newly added supplementary table to support the statement (page 28, line 19-21).

The results of the SILAC proteomic experiments (the list of the proteins down-regulated in the presence of GC7) should be made available.

We apologise that the supplementary table for the SILAC proteomics analysis was omitted by mistake. We have now included the supplementary to table (Table S5) to show all proteins changing with the corresponding fold change and p-value.

References

1. Kang, H.A. & Hershey, J.W. Effect of initiation factor eIF-5A depletion on protein synthesis and proliferation of *Saccharomyces cerevisiae*. *Journal of Biological Chemistry* **269**, 3934-3940 (1994).
2. Grosso, S. *et al.* The pathogenesis of mesothelioma is driven by a dysregulated translome. *Nature Communications* **12** (2021).
3. Nie, X. *et al.* Down-regulating overexpressed human Lon in cervical cancer suppresses cell proliferation and bioenergetics. *PLoS One* **8**, e81084 (2013).
4. Zhang, H. *et al.* Elevated mitochondrial SLC25A29 in cancer modulates metabolic status by increasing mitochondria-derived nitric oxide. *Oncogene* **37**, 2545-2558 (2018).
5. Hahm, E.R. *et al.* Withaferin A-induced apoptosis in human breast cancer cells is mediated by reactive oxygen species. *PLoS One* **6**, e23354 (2011).
6. Cheng, G. *et al.* Mitochondria-targeted vitamin E analogs inhibit breast cancer cell energy metabolism and promote cell death. *BMC Cancer* **13**, 285 (2013).
7. Hanna, D.A. *et al.* H(2)S preconditioning induces long-lived perturbations in O(2) metabolism. *Proc Natl Acad Sci U S A* **121**, e2319473121 (2024).
8. Rivas-Garcia, L. *et al.* Ultra-Small Iron Nanoparticles Target Mitochondria Inducing Autophagy, Acting on Mitochondrial DNA and Reducing Respiration. *Pharmaceutics* **13** (2021).
9. Clement, P.M., Johansson, H.E., Wolff, E.C. & Park, M.H. Differential expression of eIF5A-1 and eIF5A-2 in human cancer cells. *FEBS J* **273**, 1102-1114 (2006).
10. Elvira-Blazquez, D. *et al.* YTHDC1 m(6)A-dependent and m(6)A-independent functions converge to preserve the DNA damage response. *EMBO J* **43**, 3494-3522 (2024).
11. Jenkins, Z.A., Haag, P.G. & Johansson, H.E. Human eIF5A2 on chromosome 3q25-q27 is a phylogenetically conserved vertebrate variant of eukaryotic translation initiation factor 5A with tissue-specific expression. *Genomics* **71**, 101-109 (2001).
12. Doherty, M.K., Hammond, D.E., Clague, M.J., Gaskell, S.J. & Beynon, R.J. Turnover of the human proteome: determination of protein intracellular stability by dynamic SILAC. *J Proteome Res* **8**, 104-112 (2009).
13. Tong, M., Smeekens, J.M., Xiao, H. & Wu, R. Systematic quantification of the dynamics of newly synthesized proteins unveiling their degradation pathways in human cells. *Chem Sci* **11**, 3557-3568 (2020).
14. Thoreen, C.C. *et al.* A unifying model for mTORC1-mediated regulation of mRNA translation. *Nature* **485**, 109-113 (2012).
15. Park, Y., Reyna-Neyra, A., Philippe, L. & Thoreen, C.C. mTORC1 Balances Cellular Amino Acid Supply with Demand for Protein Synthesis through Post-transcriptional Control of ATF4. *Cell Rep* **19**, 1083-1090 (2017).
16. Knight, J.R. *et al.* Eukaryotic elongation factor 2 kinase regulates the cold stress response by slowing translation elongation. *Biochem J* **465**, 227-238 (2015).
17. Stoneley, M. *et al.* Unresolved stalled ribosome complexes restrict cell-cycle progression after genotoxic stress. *Molecular Cell* **82**, 1557-1572.e1557 (2022).
18. Guo, X. *et al.* Mitochondrial stress is relayed to the cytosol by an OMA1-DELE1-HRI pathway. *Nature* **579**, 427-432 (2020).

19. Sanchez, M. *et al.* Cross Talk between eIF2alpha and eEF2 Phosphorylation Pathways Optimizes Translational Arrest in Response to Oxidative Stress. *iScience* **20**, 466-480 (2019).
20. Fessler, E., Krumwiede, L. & Jae, L.T. DELE1 tracks perturbed protein import and processing in human mitochondria. *Nat Commun* **13**, 1853 (2022).
21. Sekine, Y. *et al.* A mitochondrial iron-responsive pathway regulated by DELE1. *Mol Cell* **83**, 2059-2076 e2056 (2023).

Response to reviewers

We were pleased that three of the reviewers were fully satisfied with our extensive revisions and additional experimentation. We have now addressed 60 individual comments from the 4 reviewers, including many additional experiments that we have carried out at considerable expense, and while strengthening paper the overall conclusions of the research remain the same.

Response to reviewer 1

Original comment 1

“It is generally believed in the field, and documented in some reports (e.g. Oliverio et al. *Amino Acids* 2014 DOI 10.1007/s00726-014-1821-0; Landau et al., *JBC* 2010 <https://doi.org/10.1074/jbc.M110.106419>), that GC7 has effects unrelated to DHPS inhibition and may directly target mitochondrial proteins. Even if lower doses and specific cell lines have been used in this work, authors have not provided evidence of the specificity of GC7 under their conditions. In Figure S3 it is shown that the knockdown of both EIF5A1 and 2 induces markers of ISR and decrease of translation rate. However, the authors have knocked down EIF5As, but not DHPS, the actual target of GC7. To demonstrate the time-dependent effect of decreased/abrogated EIF5A hypusination, the authors should use inducible DHPS knockdown and/or cells stably expressing a mutant EIF5A lacking the hypusinated lysine (i.e. EIF5AK50R) in all the experiments shown in the manuscript.”

Although Oliverio et al. *Amino Acids* 2014 and Landau et al., *JBC* 2010 are cited as evidence for GC7 having DHPS-independent impacts on mitochondrial protein expression, neither contain these data.

We directly addressed the comment above by the addition of the following data to the revised manuscript:

- A. Depletion of DHPS and DOHH reduces hypusination and enhances the phosphorylation of eIF2 α , suggesting translation initiation is inhibited upon disruption of hypusination.
- B. A different compound (CPX) that inhibits a different enzyme in the hypusination reaction (DOHH) recapitulates observations with GC7, including the inhibition of translation initiation and down regulation of mitochondrial protein expression.
- C. Respirometry studies showed that the down regulation of mitochondrial proteins and subsequent sensitivity to GC7 is linked to the reserve mitochondrial capacity.

Taken together, these additional results combined with other extensive data strongly suggest that our observations with GC7 are specific to the effect on hypusination and there is feedback between inhibition eIF5 levels/activity and the ISR.

New Comment 1. Specificity of GC7 and demonstration of DHPS-dependence

“The central claim of a time-dependent translational feedback mechanism mediated by EIF5A hypusination remains insufficiently supported. I had previously suggested a time-course analysis using inducible (e.g., doxycycline-inducible) DHPS knockdown to convincingly demonstrate that the effects observed with GC7 are indeed attributable to DHPS inhibition. Instead, the authors have employed a constitutive shRNA approach targeting both DHPS and DOHH, observing effects on initiation but not elongation.

This is not equivalent to the requested approach and fails to capture the temporal dynamics shown with GC7. I strongly recommend repeating the experiments using an inducible DHPS knockdown system and collecting time-point data that parallel the early vs. late effects seen with GC7 treatment.”

Response:

We did not use the shDHPS experimental inducible system suggested as it would not provide the data to categorically answer the reviewer's question. Using an induction knockdown system, the expression of the shRNA would be variable because even in a clonal cell population the individual cell responses differ upon induction. In addition, DHPS protein half-life has been estimated ~7 hours [<https://doi.org/10.1016/j.xhgg.2023.100206> – Fig 2], adding a further layer of complexity and, in addition to variability across the cell population, would likely result in DHPS slowly depleting over the course of 7-14 hours. As our data show the signalling from hypusination inhibition via elongation to initiation occurs within 6 hours, we would be unable to capture the temporal dynamics with this approach or clearly obtain early vs late effects, and in essence it would be comparable to the siRNA-based approach that we have already used.

We instead utilised a range of different experiments in the revised manuscript (outlined above), including DHPS+DOHH siRNA depletion to monitor the subsequent inhibition of initiation. Reanalysis of densitometry data showed the significant, time-dependent and sequential, inhibition of elongation (3 hr) followed by initiation (6 hr) which, together with polysome profiling and puromycin assays, collectively suggest inhibition of elongation by GC7 precedes the inhibition of initiation. However, we appreciate that our conclusions are based on correlative observations, and we have toned down our conclusions accordingly in the revised manuscript to take this into account.

Comment 1 cont.

“Additionally, I had proposed the use of EIF5A K50R mutants to test the role of hypusination directly. The authors claim that these mutants are unstable and cite Khang et al. (JBC 1994, Ref #34 in the manuscript and Ref #1 in the rebuttal letter) to support this. However, that study provides no data regarding K50 hypusination or EIF5A protein stability. In contrast, Sun et al. (J. Cell. Physiol. 2010; 223(3):798-809) demonstrate that WT and K50R EIF5A1 are expressed at similar steady-state levels (see their Figure 2). Therefore, the authors are still encouraged to test this mutant construct as it remains a valid and mechanistically informative tool.”

Response:

We apologise for the misciting a paper this should have been [<https://doi.org/10.1111/j.1742-4658.2007.06172.x>] which shows that human K50R in yeast is less stable than WT (Fig 2c). Inhibition of the proteasome also results in the specific accumulation of unhyposinated eIF5A [<https://doi.org/10.1038/sj.onc.1206738>] and our data show a decrease in total eIF5A following treatment with GC7, suggesting that unhyposinated eIF5A may be less stable.

In addition, Sun et al 2010 (figure 2) shows that mutant K50A (not K50R as stated) had less expression than WT eIF5A1 (below, comparison of boxed bands), suggesting that the mutant is less stable.

"Figure 2 in Sun Z, Cheng Z, Taylor CA, McConkey BJ, Thompson JE. Apoptosis induction by eIF5A1 involves activation of the intrinsic mitochondrial pathway. J Cell Physiol. 2010 Jun;223(3):798-809. doi: 10.1002/jcp.22100. PMID: 20232312. Bands of interest are in the top panel A1 72h and M 72h".

Taking these points into consideration, using a K50 mutant cell line would likely be uninformative.

Comment 2. In vivo relevance

No in vivo validation has been provided. As noted previously, animal models would be essential to assess both the pathophysiological significance and therapeutic relevance of the proposed mechanism. Without in vivo experimentation, the translational claims remain speculative.

The text has been toned down throughout the document to take into account that no animal studies have been carried out.

Comment 3

“The issue of isoform specificity has not been satisfactorily addressed. The authors assert that EIF5A1 is the predominant isoform in A549 cells and cite Elvira-Blázquez et al. (EMBO J 2024) in support. However, this reference does not provide isoform-specific expression data in A549. Contrary to the authors’ claim, other studies show that EIF5A2 is detectably expressed in A549 cells. For example, Cell Death Dis. (2022) 13:683 reports EIF5A2 protein expression in this cell line. Furthermore, EIF5A2 levels are upregulated following EIF5A1 knockdown, as shown in Cell Communication and Signaling (2023) 21:54 (see Figure 2). This suggests potential compensatory or redundant functions that have not been adequately explored. Moreover, in Figure S3A, the EIF5A1/2 blot shows only a single band. Given the distinct molecular weights of the two isoforms, a doublet should be observed. The lack of isoform discrimination raises concerns about antibody specificity or gel resolution, and should be clarified.”

Response: We apologise we were not clear about the Elvira-Blázquez et al data. The RNAseq data from this paper showed that the level of eIF5A2 expression in A549 cells was very low compared to eIF5A1.

Our analysis of publicly available RNAseq data in A549 cells (Elvira-Blázquez et al. EMBO J 2024).

Cell Death Dis. (2022) 13:683 is used to suggest that eIF5A2 is highly expressed in A549 cells. However, the materials and methods section of that paper indicate that the study used antibody ab126733, which according to the manufacturer datasheet, recognises both A1 and A2 isoforms. As the authors of this manuscript fail to determine the specificity of their antibody (i.e., there is no siRNAs for eIF5A1 and they do not fully validate the siRNAs for eIF5A2), it is unclear if they are detecting A1 or A2 protein.

We have now added additional blots (see supplementary Figure 3a) to confirm that antibody eIF5A1/2 (#17069-1-AP) recognises both A1 and A2 isoforms (only used in supplementary figure 3a to confirm protein depletion), whereas antibody eIF5A1 (#ab32443) only recognises eIF5A1 (used throughout the manuscript). These data show that eIF5A2 protein levels are very low in A549 cells. We also include RT-qPCR analysis which shows that eIF5A1 is the main isoform expressed in A549 cells (Supplementary Figure 1a) consistent with publicly available RNAseq data from A549 cells from Elvira-Blázquez et al., (above).

Although interesting, the comments made concerning eIF5A1/2 compensatory or redundant functions are beyond the scope of the paper and are not relevant in the context of our studies since i) GC7 inhibits both eIF5A1 and eIF5A2, and ii) we have used combined knockdown of eIF5A1 and 2 in all assays, which will block any compensatory increases in the other isoform. We are not studying differential roles of these highly related proteins and do not speculate or comment about this matter in the manuscript.

Reviewer #5 (Remarks to the Author):

I have carefully reviewed the revised manuscript by Sfakianos & Raven et al., along with the reviewer-1 comments and the author rebuttals. The central concern raised by Reviewer-1 relates to potential off-target effects of the GC7 compound. There is indeed some literature describing eIF5A-independent cellular effects of GC7 (e.g. on autophagy - Oliverio et al., 2014), and as with most small-molecule inhibitors, some degree of off-target activity is not unexpected (e.g. for instance, even some of our most specific kinase inhibitors used in clinic hit dozens of other targets). The critical question, however, is whether the molecular and cellular effects of GC7 reported here can be confidently attributed to inhibition of eIF5A.

To address this point, the authors present several independent lines of evidence:

- 1- RNAi-mediated depletion of eIF5A reproduces the effects of GC7.
- 2- A distinct small-molecule inhibitor of the pathway (CPX), which targets DOHH rather than DHPS (target of GC7), also recapitulates the GC7 effects.
- 3- RNAi-mediated depletion of DOHH and DHPS similarly phenocopies GC7 treatment.
- 4- The observed effects can be rescued by ISRIB, demonstrating that the downstream pathway identified here is essential for the cellular phenotype.
- 5- A rescue is also achieved by mutating eIF2 α on S51, the inhibitory phosphorylation site mediating the initiation block downstream of eIF5A.

It would be highly improbable for the experiments 1, 2, & 3 to yield concordant results solely due to off-target effects. As a result, these findings strongly support the conclusion that the cellular effects of GC7 are mediated via inhibition of eIF5A function. Furthermore, experiments 4 and 5 provide compelling evidence of functional epistasis between the the translation-initiation regulation described here and the phenotypic consequences of GC7 treatment (as opposed to other potential off-targets effects of this compound), further strengthening the conclusions of the study.

Taken together, the presented evidence in my view is very convincing. While I fully appreciate the initial concerns of the reviewer-1 regarding GC7 specificity, the breadth of additional supporting data, using multiple different means of targeting the same pathway, substantially mitigates this issue. I therefore recommend that the manuscript is accepted for publication. Importantly, I do not believe the inducible knockdown experiment that the reviewer is suggesting in their latest comments will

add anything further, as RNAi still takes a while to significantly deplete a protein target (inducible or not), so the result of this experiment will be basically the same as what is already shown in terms of RNAi-mediated eIF5A depletion.

My only minor suggestion to the authors is that in the Discussion section they may want to acknowledge the potential for GC7 having off-target effects, but also clearly articulate why their experimental results argue strongly against this being a concern. Summarising the key supporting lines of evidence mentioned above would pre-empt similar concerns from future readers and would further strengthen the manuscript in my opinion.

Reviewer #5 (Remarks on code availability):

N/A

Author response

We would like to thank the reviewer for their assessment of our manuscript, and we are pleased that they agree with reviewer 2, 3 and 4 in deeming our revised manuscript is suitable for publication.

As suggested, we have added a paragraph to the discussion highlighting potential off-target effects of GC7 and explained how our experimental approach has minimised these possibilities.

Reviewer #1 (Remarks to the Author)

I appreciate the effort made in revising the manuscript. However, the key concerns remain unresolved, particularly regarding the specificity of the experimental system and the robustness of the mechanistic conclusions.

The central claim — that inhibition of elongation through reduced eIF5A activity leads to feedback inhibition of translation initiation — is conceptually interesting. However, this model is based entirely on experiments using GC7, a compound with known and emerging off-target effects. In particular, data presented at the 2024 International Conference on the Biological Roles of Polyamines (Kobe, Japan) reported that GC7 can inhibit mitochondrial complex I, which is known to induce energetic stress, activate the integrated stress response (ISR) via eIF2 α phosphorylation, and lead to global inhibition of translation initiation. This mechanism closely mimics the effects described in the manuscript, raising the possibility that the observed phenotype is not mediated by inhibition of eIF5A hypusination, but rather by indirect activation of ISR due to mitochondrial dysfunction.

These concerns are reinforced by observations from our group, which has also noted off-target effects of GC7 in unrelated experimental contexts, consistent with the possibility that the observed translation phenotypes may not be solely due to inhibition of eIF5A hypusination.

Author response

The data highlighted by the reviewer is of course interesting, but it is non-peer reviewed and unpublished data (i.e., unclear what model system/techniques/concentration of GC7 was used), and we are therefore unable to access it to take it into consideration. However, our data utilising the glu/gal model suggests GC7 is not a direct ETC inhibitor as ATP does not decrease until >6 hr treatment in galactose (supplementary figure 10a), whereas direct ETC inhibitors would typically reduce ATP production in galactose within 1 hr. In addition, ATP is reduced in MCF10A to the same extent in glucose and galactose (supplementary figure 10a), again suggesting GC7 is not a direct ETC inhibitor.

We have amended the text to make clear that the data suggests GC7, at least at the concentrations used within our study, is not a direct ETC inhibitor.

This issue is particularly critical because the temporal sequence proposed by the authors — early inhibition of elongation (3 h), followed by delayed inhibition of initiation — is demonstrated exclusively using GC7. Since this biphasic response is a central conceptual point of the manuscript, the lack of orthogonal validation substantially limits the strength and specificity of the conclusions.

While the authors argue that genetic approaches may not fully recapitulate the timing of pharmacological inhibition, this does not justify their exclusion. At least two complementary strategies are both feasible and necessary:

An inducible DHPS knockdown or knockout system, enabling controlled, time-resolved hypusination inhibition while avoiding off-target drug effects.

A DHPS knockout cell line rescued with a non-hypusinable eIF5A mutant (e.g., K50A or K50R). Although the authors mention potential instability, this is not demonstrated experimentally. Titration of the expression vector could mitigate this issue. The immunoblot shown at 72 h is not informative in this regard and is not directly comparable to earlier time points (e.g., 48 h).

Author response

As detailed in our previous response, we have utilised a range of independent approaches (depletion of eIF5A, depletion of DHPS and DOHH hypusination enzymes, utilising a distinct inhibitor of DOHH, rather than DHPS that is inhibited by

GC7), that all recapitulate our observations with GC7, and it is highly improbable that all these approaches would yield the same off-target effect independently.

We did previously acknowledge the potential off-target activity of GC7 in the manuscript, but we have now included a paragraph in the discussion expanding this further, as well as presenting the rationale for our approach to mitigate these risks.

The possibility that protein stability and response dynamics (e.g., DHPS, eIF5A) vary across cell types is another relevant factor that is disregarded by the authors.

Author response

We make no claims about the conservation of this response and explain our rationale for focussing on lung (A549) in the introduction. We observe similar trends in cell lines derived from lung, mesothelial, breast, cervical and colorectal tissue, suggesting that this response may be conserved, however, we were careful not to over interpret our data as this would be inappropriate without further investigation.